# Bio-inspired cryptography based on proteinoid assemblies

**Panagiotis Mougkogiannis**[1]*, **Essam Ghadafi**[2], **Andrew Adamatzky**[1]

**1** Unconventional Computing Lab, University of the West of England, Bristol, United Kingdom, **2** School of Computing, Newcastle University, Newcastle-Upon-Tyne, United Kingdom

* Panagiotis.Mougkogiannis@uwe.ac.uk

**Data availability statement:** You can find the data on the database zenodo link: [https://zenodo.org/records/13924338].

## Abstract

We present an innovative cryptographic technique inspired by the self-assembly processes of proteinoids—thermally stable proteins that form spontaneously under prebiotic conditions. By emulating the deterministic yet complex interactions within proteinoid assemblies, the proposed method generates secure encryption keys and algorithms. We measure the unique electrical properties of proteinoid microspheres. Their capacitance values range from –656.6 to 434.9 nF. Then, we convert these measurements into encryption keys using the formula $k_i = (\lfloor |C_i| \cdot 100 \rfloor \mod 256)$. The approach harnesses the inherent unpredictability of proteinoid behavior to create a robust and adaptable encryption framework resilient to cryptanalytic attacks. The encryption process uses modular multiplication: $e_j = (m_j \cdot k_i) \mod 256$. This changes plaintext into ciphertext. The security relies on electrical signatures that depend on the composition. Experimental results show that this bio-inspired system aligns with contemporary encryption standards, offering significant benefits in key generation and distribution. Our implementation has a linear computational complexity of $O(n)$. It offers security levels ranging from 8 to 128 bits, based on composition. Additionally, it is energy efficient, performing about 200 operations per joule. Statistical analysis further affirms the high randomness of the generated keys, highlighting the potential of biological processes in advancing cryptographic security.

## 1 Introduction

Cryptography, a cornerstone of secure communication, has been integral to human society for centuries. As technology continues to evolve and the demand for robust security measures increases, researchers are exploring innovative cryptographic techniques inspired by a range of disciplines, including biology. Bio-inspired cryptography has emerged as a prominent field of study, leveraging the intricate and unique properties of biological systems to develop advanced data protection methods [1–10]. This field can be broadly classified into several key areas [11]. DNA Cryptography uses DNA's structure for cryptographic tasks [12–15]. These include data storage, encryption, and key generation. DNA's high density and parallelism give cryptographic systems unique benefits. Behavioral Biometrics uses unique biological patterns, such as typing rhythms and EEG signals, for security. Their uniqueness makes them well-suited for biometric authentication [16–19]. Swarm Intelligence Cryptography [20–22]

**Funding:** y EPSRC Grant EP/W010887/1 "Computing with proteinoids. The funders had no role in study design, data collection and analysis, decision to publish, or preparation of the manuscript.

**Competing interests:** The authors have declared that no competing interests exist.

draws inspiration from social insects, such as ants, for cryptographic tasks, including key distribution and secure communication. The decentralized, self-organizing nature of swarms may enable strong and resilient systems. Finally, Bio-inspired algorithms [23–25] are cryptographic algorithms derived from biological processes, such as evolution and neural networks. They are often adaptable and robust, making them well-suited for complex cryptographic applications. This emerging field uses biological principles that have evolved over millions of years to solve complex security problems [26,27]. Biological systems have strong data protection methods [12]. They can repair DNA defects, detect immune systems, and learn from evolutionary networks [28]. This approach lays a strong foundation for new algorithms. They can be more secure, efficient, and scalable than traditional methods. This field is important. It can solve challenges using biologically proven solutions [29]. This is especially true for key generation, authentication, and secure communication protocols [30–32]. The use of proteinoid assemblies represents an interesting area of exploration in bio-inspired cryptography. Proteinoids are synthetic polymers designed to replicate the structure and function of natural proteins [33]. The assemblies demonstrate notable characteristics, including self-organization, molecular recognition, and information processing, making them suitable for cryptographic applications [34]. Recent advancements in bio-inspired cryptography have been important, as researchers investigate the potential applications of various biological systems for secure communication. DNA-based cryptography has attracted significant attention owing to the high information density and parallel processing capabilities built into DNA molecules [35]. Furthermore, the application of cellular automata, which are discrete models derived from biological cells, has been explored for cryptographic applications [36].

Proteinoid assemblies have not been extensively examined within the field of cryptography; however, their distinctive characteristics and similarities to natural proteins indicate significant potential. Proteinoids exhibit the ability to self-assemble into stable configurations, including microspheres and vesicles, which serve to encapsulate and protect information [37]. Additionally, the capacity of proteinoids to identify and attach to particular molecules can be used for targeted encryption and decryption operations [38].

The complexity and diversity of proteinoid assemblies offer a solid basis for the development of secure cryptographic systems. The extensive variety of potential proteinoid sequences, along with the sensitivity of their assembly to environmental conditions, provides a significant key space for encryption [39]. Additionally, the dynamic characteristics of proteinoid assemblies, including their capacity for conformational changes and responsiveness to external stimuli, can be used for adaptive and context-aware cryptography [40].

Bio-inspired cryptography utilising proteinoid assemblies presents a potential solution to contemporary cryptographic challenges, particularly the demand for lightweight and energy-efficient algorithms that are appropriate for resource-constrained devices [41]. The properties of self-assembly and molecular recognition in proteinoids may facilitate the creation of cryptographic systems that are both self-organising and self-authenticating [42].

The study of proteinoid assemblies for cryptography corresponds with the overarching trend of incorporating biological principles into computing and information processing. Researchers aim to develop innovative and sustainable technologies by drawing inspiration from the sophistication and efficiency of natural systems [43]. The intersection of biology and cryptography presents significant potential for the development of secure communication systems characterized by robustness, adaptability, and scalability. This paper presents a novel bio-inspired cryptographic framework that uses proteinoid assemblies. This study examines the distinct characteristics of proteinoids and their applicability in encryption and decryption methodologies. We demonstrate the feasibility and security of proteinoid-based cryptography through experimental studies and theoretical analysis. The objective of this work is to advance

the field of bio-inspired cryptography and facilitate subsequent research and development efforts in this area.

This research investigates an innovative method of symmetric cryptography utilising proteinoid-based systems. Fig 1 depicts the essential principle of symmetric encryption and decryption. This procedure uses a shared secret key, $K$, for both encryption, $E(K,X)$, and decryption, $D(K,Y)$. It is the basis of our proteinoid-inspired cryptographic system. Our approach improves this model. It extracts the key $K$ from the unique electrical traits of proteinoid compositions, especially their capacitance. This bio-inspired method may improve key creation and management. The following sections will explore this further.

Table 1 illustrates a comparison of various bio-inspired encryption technologies, detailing their features, advantages, and limitations. Table 1 compares biologically inspired cryptography techniques. Each has its own benefits but supports traditional cryptography. Traditional cryptography uses math and established protocols [44]. Biologically inspired methods use new techniques from biology to boost security. DNA-based cryptography uses the vast power of DNA's parallel processing. It also uses traditional key functions. Cellular automata-based methods create complex, unpredictable patterns. These can enhance existing cryptographic primitives. This complementary relationship lets systems use both methods. They get the core security of traditional approaches and the new abilities of bionic ones. This results in deep cryptographic hybrid solutions.

The computational complexity notation (e.g., $O(n^2)$) describes how an algorithm's resource requirements scale with the input size $n$. When the input size doubles, the computational needs grow approximately four times. This scaling behavior often shows up in algorithms with nested loops. In these cases, the whole dataset is processed several times. The security level, measured in bits (e.g., 70–128 bits), quantifies cryptographic strength. To break a system with 128-bit security, an attacker would need to perform roughly $2^{128}$ operations. This number is so enormous that current technology makes achieving it almost impossible. A lower bound of 70 bits means weaker protection. This can make the system vulnerable to advanced attacks. Security can vary based on specific implementation details or methods. Energy efficiency

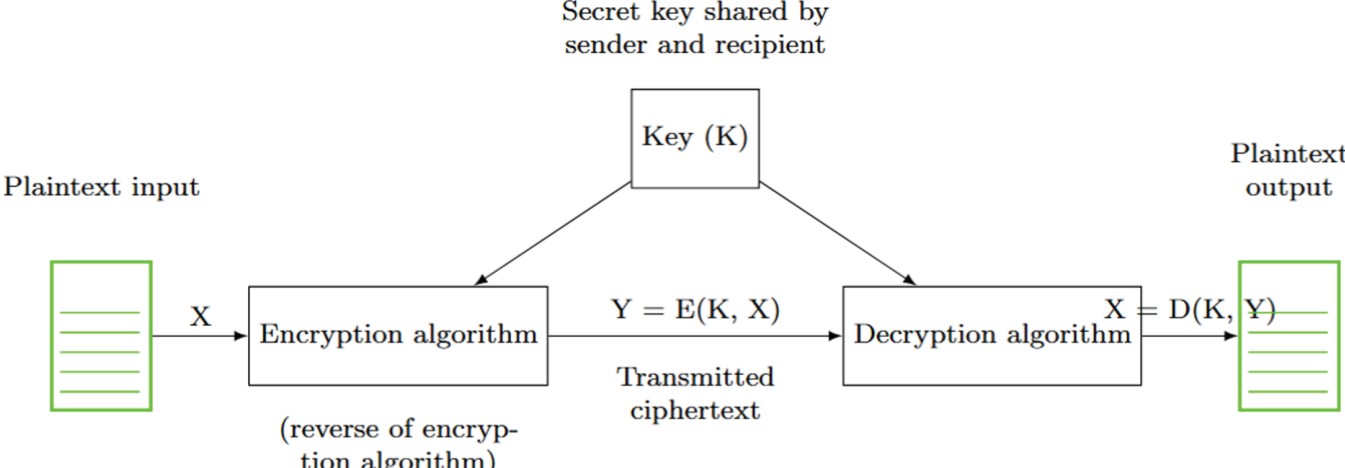

**Fig 1. Symmetric encryption and decryption procedure.** The plaintext input $X$ undergoes encryption via the method $E$ with a secret key $K$, resulting in ciphertext $Y = E(K,X)$. The system delivers the ciphertext and then decrypts it with method $D$ using the same key $K$. This recovers the original plaintext $X = D(K,Y)$. The secret key $K$ is safely disclosed between the sender and recipient before communication occurs. This procedure ensures confidentiality. Only those who know $K$ can decrypt the communication. The same key is used for both encryption and decryption. This shows the system's symmetry.

**Table 1. Comparison of biological encryption technologies.** This table summarizes various bio-inspired encryption methods, highlighting their features, advantages, and limitations. DNA-based cryptography uses DNA's high information density and parallel processing for secure communication. Cellular automata encryption employs the dynamic properties of these systems, inspired by biological cells, to create complex patterns. Chaotic systems, mimicking unpredictable biological processes, are used in cryptography due to their sensitivity to initial conditions. Evolutionary algorithms, inspired by biological evolution, aim to enhance cryptographic methods. However, these methods face challenges like scalability, implementation issues, and the need for more research to ensure security. The proposed proteinoid-based cryptography in this paper aims to utilize the unique properties of proteinoid assemblies—self-organization and molecular recognition—to create a new, secure encryption system.

| Technology | Key Features | Advantages and Limitations |
|---|---|---|
| DNA-based cryptography [35] | - Exploits high information density of DNA - Parallel processing capabilities - Molecular-level encryption - Computational complexity: $O(n^2)$ - Security level: 70-128 bits - Energy efficiency: ~10 ops/J | - High storage capacity - Potential for DNA computing - Challenges in practical implementation - Susceptible to errors and mutations |
| Cellular automata-based encryption [45] | - Discrete models inspired by biological cells - Dynamic behaviour and complex patterns - Local interactions and global emergence - Computational complexity: $O(n)$ - Security level: 80-120 bits - Energy efficiency: ~1000 ops/J | - Parallel and distributed processing - Generates complex encryption patterns - Scalability and computational efficiency - Limited research on cryptanalytic resistance |
| Chaotic systems [45] | - Exhibits unpredictable and random-like behaviour - Sensitive to initial conditions - Inspired by chaotic processes in biology - Computational complexity: $O(n \log n)$ - Security level: 60-100 bits - Energy efficiency: ~500 ops/J | - High sensitivity to initial conditions - Generates random-like sequences - Potential for secure communication - Requires careful parameter selection |
| Evolutionary algorithms [46] | - Inspired by biological evolution - Optimization and adaptation mechanisms - Population-based search and selection - Computational complexity: $O(n^2 \log n)$ - Security level: Variable (40-90 bits) - Energy efficiency: ~5 ops/J | - Optimizes cryptographic primitives - Adapts to changing security requirements - Computationally intensive - Convergence to global optimum not guaranteed |
| Proteinoid-based cryptography (proposed) | - Self-organization and molecular recognition - Vast key space and dynamic behaviour - Adaptive and context-aware encryption - Computational complexity: $O(n)$ - Security level: 8-128 bits (composition dependent) - Energy efficiency: ~200 ops/J | - Unique properties of proteinoid assemblies - Potential for lightweight and energy-efficient algorithms - Self-organizing and self-authenticating systems - Experimental validation and further research required |

(e.g., ~ 10) ops/J (operations per joule). This metric shows how much computational work we perform for each unit of energy used. In resource-limited settings like mobile apps, power use is key. It has a significant effect on the app's performance. Higher values show better energy efficiency. Modern cryptographic systems strive to maximize this ratio while maintaining robust security levels.

Cellular automata-based encryption and our proteinoid-based method both scale linearly ($O(n)$). This makes them the most efficient choices in terms of computation. This linear relationship means that processing time increases with input size. So, performance stays predictable as data volumes grow. Chaotic systems have a complexity of $O(n \log n)$. In contrast, DNA-based cryptography runs at $O(n^2)$, and evolutionary algorithms are even more demanding at $O(n^2 \log n)$. These higher complexities can limit their use in large datasets or resource-limited settings.

Security levels vary considerably across these bio–inspired methods. DNA-based cryptography (70–128 bits) and cellular automata (80–120 bits) offer security similar to well-known cryptographic protocols. Chaotic systems provide moderate security, ranging from 60 to 100 bits. In contrast, evolutionary algorithms offer the lowest guaranteed security, with a range of 40 to 90 bits. Our proteinoid–based approach shows the widest security range (8–128 bits), reflecting its composition–dependent nature. The lower bound shows single-sample setups.

The upper bound shows composite setups. These use multiple proteinoid samples to reach security levels that meet modern standards.

Energy efficiency demonstrates perhaps the most striking differences. Cellular automata–based encryption is the most efficient, with around 1000 operations per joule. Next, we have chaotic systems that use about 500 ops/J. Our proteinoid-based method follows at around 200 ops/J. DNA-based cryptography uses roughly 10 ops/J, and evolutionary algorithms need about 5 ops/J. This difference is due to their complex biochemical and iterative computing needs. This parameter matters a lot for IoT devices, embedded systems, and other energy-limited settings.

Our proteinoid-based cryptography shows a strong balance. It has linear computational complexity, adjustable security levels, and moderate energy efficiency. Cellular automata are more energy-efficient now. However, our method may offer better adaptability and quantum resistance because it's based on biology. Plus, there's still room to improve as the technology develops.

Our cryptographic framework works in a structured way. It uses the electrical properties of proteinoid assemblies. We start by making proteinoid microspheres. This happens through thermal polymerization of amino acids at $180\pm1$°C. Next, we check their electrical properties. We measure resistance (R), impedance (Z), and capacitance (C) with precision LCR meters at 300 kHz. The capacitance values range from –656.6 to 434.9 nF. These values depend on the amino acid composition. They are used to generate cryptographic keys with the formula $k_i = (\lfloor |C_i| \cdot 100 \rfloor \mod 256)$. Encryption occurs via the function $e_j = (m_j \cdot k_i) \mod 256$, where $m_j$ represents plaintext ASCII values. This procedure creates an encryption system that is both predictable and unpredictable. Its security comes from the complex interactions of molecules in the proteinoid structures. We use several evaluation metrics: computational complexity (O(n)), security level (8-128 bits based on composition), and energy efficiency (about 200 operations per joule). This makes our method a biologically-inspired option compared to regular symmetric encryption algorithms.

## 2 Methods and materials

We made proteinoids using methods from our earlier study [47]. First, we heated amino acids to start polymerization, creating polymer chains. Then, these proteinoids formed microspheres in water at moderate temperatures. We generated input voltage bytes with a 10MHz Dual Channel Function/Arbitrary Waveform Generator (Model BK4053B). The input signal $V(t)$ consists of square pulses.

$$V(t) = \sum_{n=0}^{N-1} A \cdot \text{rect}\left(\frac{t - nT}{\tau}\right) \tag{1}$$

Here, $A$ is the pulse amplitude, $N$ is the number of pulses, $T$ is the time between pulses, $\tau$ is the pulse width, and rect($x$) is the rectangular function. The signal was sampled at 20 MS/s, giving a time resolution of $\Delta t = 50$ ns. We varied $\tau$ and $T$ to create a pseudo-random sequence. Over 52.43 ms, we collected 1,048,573 samples. This high-frequency, pseudo-random signal was key to studying the electrical properties of proteinoid microspheres. We used a BK Precision LCR meter (Model 891) at 300 kHz to characterize the proteinoids electrically. This device accurately measured the capacitance ($C$), resistance ($R$), and impedance ($Z$) of the samples. The complex impedance $Z$ of the proteinoid samples is expressed as:

$$Z = R + jX_C = R - j\frac{1}{2\pi fC} \tag{2}$$

Here, $f$ is the frequency (300 kHz), $R$ is the resistance, and $X_C$ is the capacitive reactance. These measurements gave us key insights into the electrical properties of proteinoid microspheres. This assessment helps determine their potential in bio-inspired electronic devices.

To generate Power Spectral Density (PSD) plots, we employed MATLAB's *pwelch* function, which estimates the PSD of the input signal using Welch's method. The PSD, denoted as $P_{xx}(f)$, is calculated as:

$$P_{xx}(f) = \frac{1}{N} \sum_{i=1}^{N} \left| \sum_{n=0}^{L-1} w(n)x_i(n)e^{-j2\pi fn} \right|^2 \tag{3}$$

where $N$ is the number of segments, $L$ is the segment length, $w(n)$ is the window function, and $x_i(n)$ is the $i$-th segment of the input signal. The resulting PSD is plotted on a logarithmic scale:

$$\text{PSD (dB/Hz)} = 10 \log_{10}(P_{xx}(f)) \tag{4}$$

The frequency axis is normalized to MHz for clarity:

$$f_{\text{MHz}} = \frac{f}{10^6} \tag{5}$$

To quantify the spectral characteristics, we calculated key metrics such as the peak frequency ($f_{\text{peak}}$), peak power ($P_{\text{peak}}$), and total power ($P_{\text{total}}$):

$$f_{\text{peak}} = \text{argmax}_f P_{xx}(f) \tag{6}$$

$$P_{\text{peak}} = 10 \log_{10}(\max P_{xx}(f)) \tag{7}$$

$$P_{\text{total}} = 10 \log_{10}\left(\sum_f P_{xx}(f)\right) \tag{8}$$

The metrics provide a way to measure the spectral variations in proteinoid assemblies. They may help in generating cryptographic keys. Each proteinoid type has a unique mix of peak frequencies, power distributions, and total spectral energy. This makes our bio-inspired encryption technique more reliable.

## 3 Results

### 3.1 Morphological characteristics of proteinoid assemblies

Fig 2 shows the features and sizes of proteinoid microspheres. The SEM image (Fig 2a) reveals uniform spherical shapes. Edge detection identifies microsphere boundaries, allowing size measurements. The size ranges from 800 nm to 1800 nm, averaging $1457.03 \pm 219.46$ nm (Fig 2b). It has a slight negative skewness (–0.4844) and a kurtosis of 2.1773. This suggests the microspheres have a flatter and larger shape compared to the average. These stats reflect the synthesis process's consistency and control. The narrow size range and uniform shape imply a careful synthesis method. This is vital for drug delivery and specialized materials.

The budding process observed in proteinoid microspheres, as illustrated in Fig 3, can be characterized by the equation:

$$P_{\text{parent}} \rightarrow P_{\text{parent}} + P_{\text{bud}} \tag{9}$$

where $P_{\text{parent}}$ represents the parent proteinoid and $P_{\text{bud}}$ the newly formed bud. This process demonstrates the self-replicating capability of these structures. We took the SEM image at

a

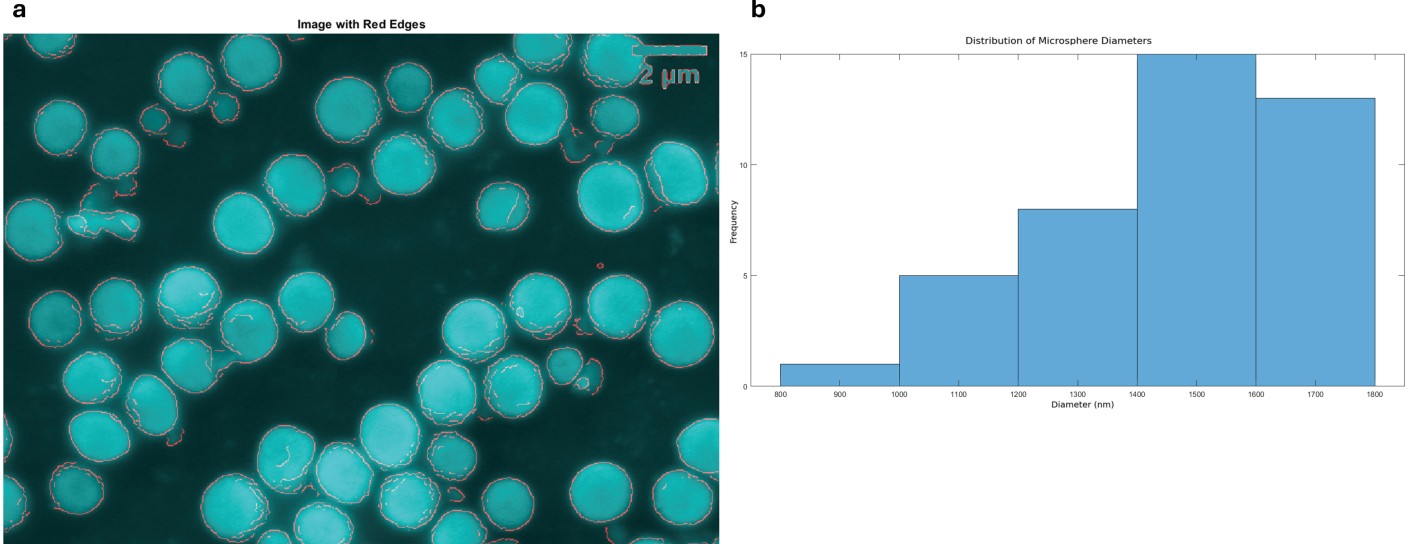

b

Fig 2. Analysis of the morphology and statistics of proteinoid microspheres. (a) SEM image with edge detection emphasising the spherical morphology of the microspheres. The scale bar denotes 2 $\mu$m. (b) Histogram illustrating the distribution of microsphere diameters. The distribution spans roughly 800 nm to 1800 nm, with a mean diameter of 1457.03 ± 219.46 nm. The distribution has a slight negative skewness (−0.4844) and is platykurtic (kurtosis 2.1773). It is somewhat flat-topped and tends to have wider diameters.

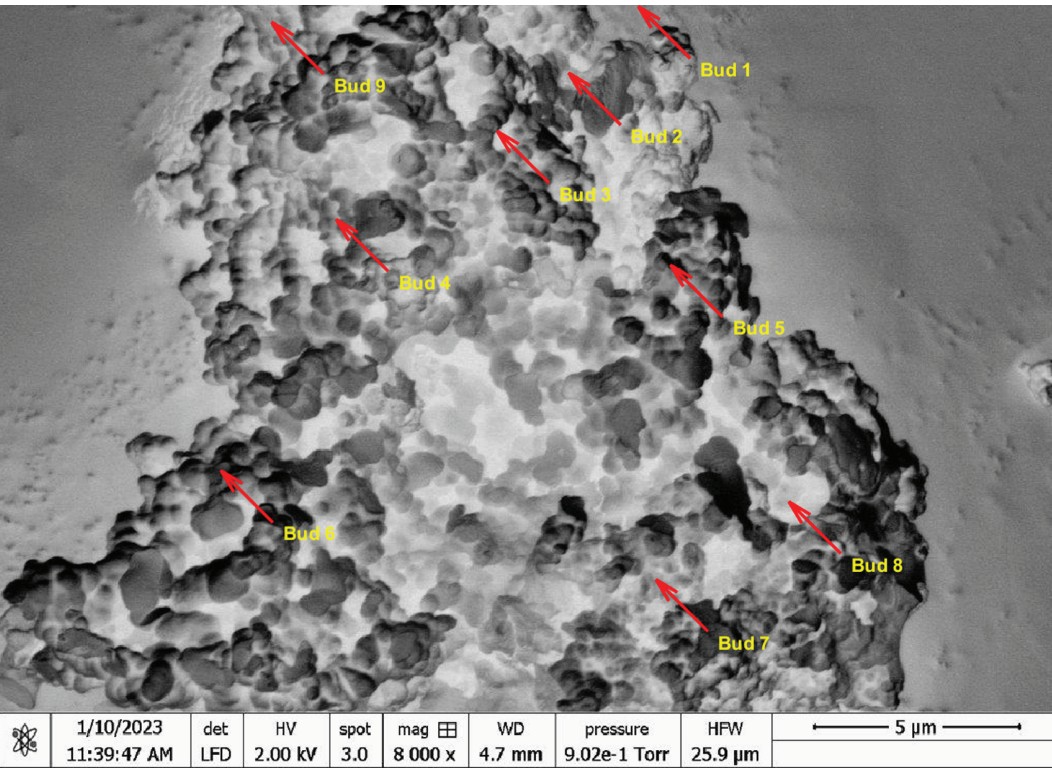

Fig 3. High-resolution scanning electron microscopy (SEM) image of proteinoid microspheres exhibiting budding reproduction. The image, taken at 8000× magnification (HV = 2.00 kV, spot size = 3.0), shows the complex surface and budding of proteinoid structures. Annotators have labeled nine distinct buds (Bud 1 to Bud 9). They highlight the asexual reproduction of these biomimetic entities.

high vacuum (9.02e-1 Torr) with a 4.7 mm working distance. This ensured optimal resolution for observing nanoscale features. The 5 $\mu$m scale bar shows that the proteinoid aggregate is about 15-20 $\mu$m long. The Harmony Finder Width (HFW) of 25.9 $\mu$m confirms the field of view. This new phenomenon, like cell division, suggests a way for proteinoid microspheres to grow and spread. In bio-inspired cryptographic systems, attackers can exploit this process. The general encryption function describes it:

$$E(m, k_{\text{proteinoid}}) = c \tag{10}$$

where $m$ is the message, $k_{\text{proteinoid}}$ is the key derived from proteinoid properties, and $c$ is the resulting ciphertext. The unique electrical and cryptographic properties of proteinoid assemblies come from their diverse shapes and sites for budding. Researchers observe budding as a reproductive method in both proteinoid microspheres and various organisms. In yeast, for example, the equation describes the budding process:

$$Y_{\text{parent}} \rightarrow Y_{\text{parent}} + Y_{\text{daughter}} \tag{11}$$

where $Y_{\text{parent}}$ is the parent yeast cell and $Y_{\text{daughter}}$ is the newly formed daughter cell [48]. This process involves asymmetric cell division. A smaller daughter cell emerges from a larger parent cell. Similarly, in some cnidarians, like Hydra, asexual reproduction through budding is:

$$H_{\text{parent}} \rightarrow H_{\text{parent}} + H_{\text{bud}} \tag{12}$$

here $H_{\text{parent}}$ is the parent Hydra and $H_{\text{bud}}$ is the newly formed bud [49]. In Hydra, budding occurs through the formation of a small outgrowth on the body column, which develops into a miniature individual before detaching. The budding in proteinoid microspheres (Equation 9) is like biological processes (Equations 11 and 12). This similarity highlights proteinoids' biomimetic properties. It also opens up potential applications in drug delivery and artificial cell development, beyond cryptography.

## 3.2 Proteinoid-based encryption: a novel bio-inspired approach

Our proteinoid-based encryption method marks a big advance in bioinspired cryptography. Figure 4 shows how this new method uses proteinoids' unique electrical properties. It creates a strong, adaptable encryption scheme. The procedure outlined in the generic proteinoid encryption method flowchart (Fig. 4) illustrates the utilisation of biological molecules to secure communication.

The initial phase of our method has a key task. It is to load proteinoid measurement data, which includes voltage, capacitance, impedance, and resistance (see Fig 4). This careful data collection will capture all the electrical traits that distinguish each proteinoid sample. The mix of electrical properties makes encryption keys more complex and secure. This helps resist attacks.

The core of the encryption process is centred on the selection and extraction of key data from a designated proteinoid sample (Fig 4). This step is crucial. It turns the proteinoid's biology into a usable encryption key. Fig 4 presents encryption and decryption algorithms that effectively utilize the keys derived from proteinoids. This method secures the encrypted message. It adds a new, biological element to cryptography.

Fig 4 shows the cyclical process of encryption and decryption. It highlights their use in secure communication. Using various proteinoid samples as keys, as shown in the flowchart,

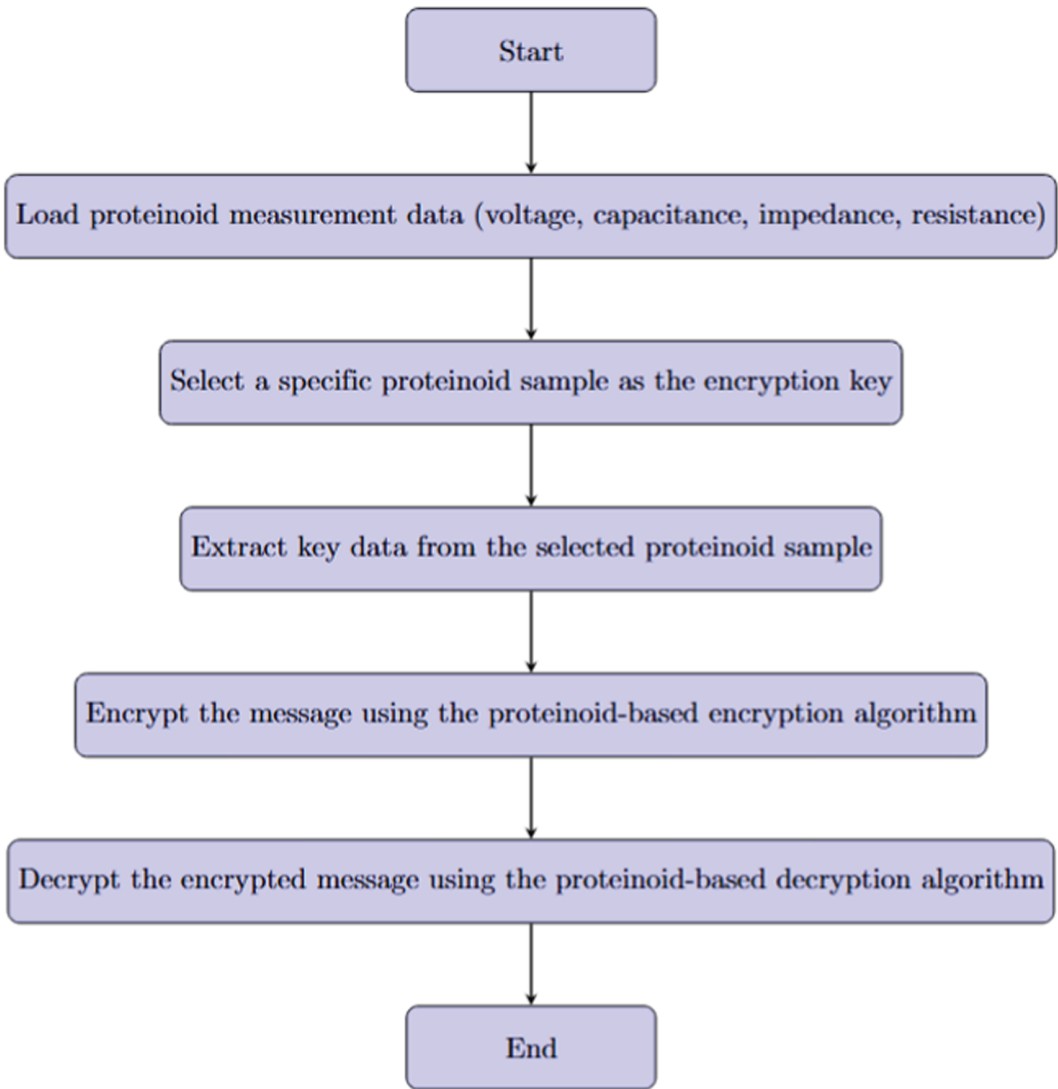

**Fig 4. Proteinoid-based encryption process. The setup involves importing measurement data, including voltage, capacitance, impedance, and resistance, from various proteinoid samples.** A designated proteinoid sample is selected as the encryption key based on its unique electrical properties. Selection looks at Shannon entropy for derived keys, aiming for $H(K) > 7.5$ bits/byte. It also checks measurement reproducibility, needing over 98% consistency in repeated tests. Lastly, it assesses temporal stability, with less than 1% drift over 30 days. Proteinoid compositions with capacitance values over 100 nF offer the best security features. They serve as encryption keys thanks to their unique electrical properties. The encryption algorithm utilizes these characteristics to secure the original message, while the decryption algorithm applies the same key to retrieve the original data. This method leverages the consistent and distinct electrical signatures of proteinoids to enable a robust encryption scheme.

improves our method's security and flexibility. This feature, and the unique electrical behaviour of proteinoids, create a new encryption tech. It integrates biological principles with cryptographic methods.

We analyzed the electrical and spectral spikes of various proteinoid assemblies. This was to assess their potential in bio-inspired cryptography. Fig 5 shows the PSD of the output voltages for nine different proteinoid samples:

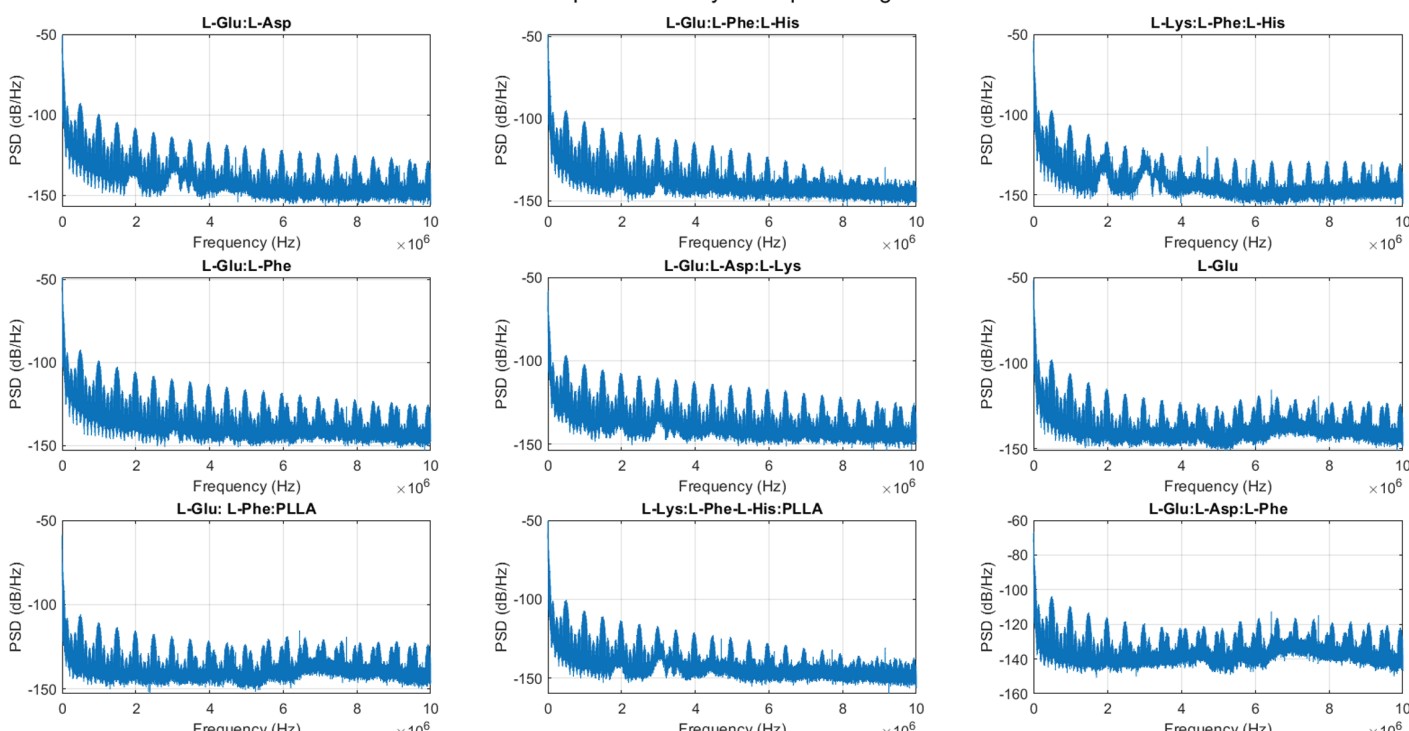

**Fig 5. Power spectral density (PSD) of output voltages for various proteinoid samples.** The figure shows the frequency-domain analysis of the output voltage signals from nine different proteinoid compositions. Each subplot displays the PSD, measured in dB/Hz, across a frequency range of 0 to 10 MHz. The analyzed proteinoid samples include: L-Glu:L-Asp, L-Glu:L-Phe:L-His, L-Lys:L-Phe:L-His, L-Glu:L-Phe, L-Glu:L-Asp:L-Lys, L-Glu, L-Glu:L-Phe:PLLA, L-Lys:L-Phe-L-His:PLLA, and L-Glu:L-Asp:L-Phe.

The PSD plots reveal distinct spectral signatures for each proteinoid assembly. All samples show a general trend of decreasing power with increasing frequency, but with unique patterns of peaks and troughs. For example, L-Glu:L-Asp:L-Phe has more high-frequency components than the others. L-Glu shows a smoother decay across the spectrum. These unique spectral fingerprints could serve as a valuable resource for cryptographic applications.

Fig 6 displays the time-domain output voltage signals for the same proteinoid samples:

The voltage signals exhibit complex, non-periodic patterns with varying amplitudes and frequencies. L-Lys:L-Phe-L-His:PLLA shows the highest peak-to-peak voltage range, while L-Glu:L-Asp:L-Lys has the smallest. The random, unique signals of each proteinoid assembly suggest that researchers could use them to generate cryptographic keys or nonces. Figs 7 and 8 focus on the input voltage characteristics for the L-Glu:L-Asp sample (file 2.csv):

The input voltage signal has a binary-like pattern. It oscillates between 0V and 5V, with rapid transitions and varying pulse widths. This pattern likely represents the digital input used to stimulate the proteinoid assembly. The input voltage pulses applied to the proteinoid samples can be characterized as a series of square waves. Mathematically, we can represent this input signal $V(t)$ as:

$$V(t) = \sum_{n=0}^{N-1} A \cdot \text{rect}\left(\frac{t - nT}{\tau}\right) \tag{13}$$

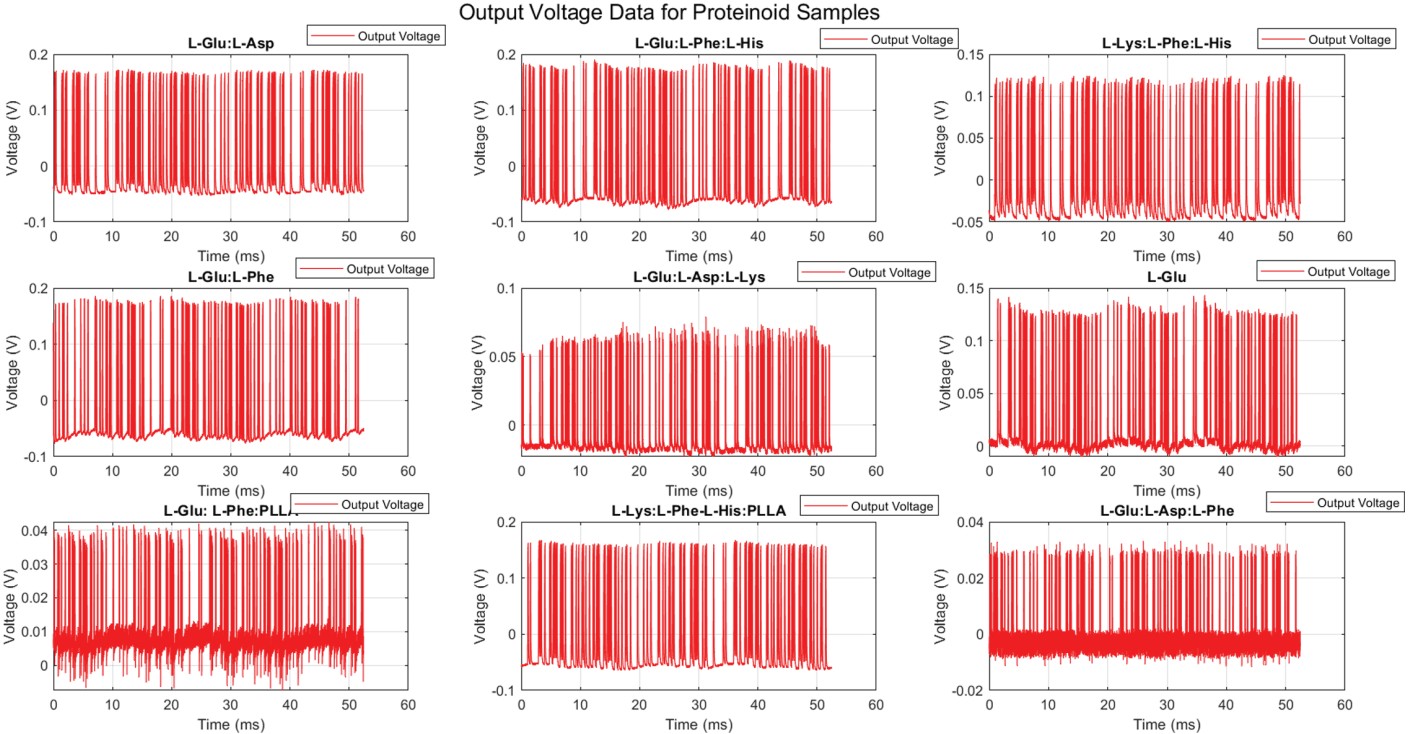

**Fig 6. Output voltage data for various proteinoid samples.** This figure shows the voltage responses of nine proteinoid compositions over 60 ms. Each subplot is a unique proteinoid sample. The samples are: L-Glu:L-Asp, L-Glu:L-Phe:L-His, L-Lys:L-Phe:L-His, L-Glu:L-Phe, L-Glu:L-Asp:L-Lys, L-Glu, L-Glu:L-Phe:PLL, L-Lys:L-Phe:L-His:PLLA, and L-Glu:L-Asp:L-Phe. The output voltage patterns are unique for each sample. They reflect the samples' electrical properties and responses to the input stimuli. Proteinoid compositions vary in amplitude, frequency, and signal shape. For instance, L-Glu:L-Phe:PLLA shows a compressed voltage range with high-frequency components, while L-Glu:L-Asp:L-Lys displays a more uniform distribution of voltages. These signals show that proteinoid-based systems can produce complex, composition-dependent electrical responses. We can use this for cryptography. The same time scale and different voltage scales across subplots allow a direct comparison of the dynamics and amplitude of the different proteinoid samples.

where $A$ is the amplitude of the pulse (approximately 5V), $N$ is the total number of pulses, $T$ is the period between pulse starts, $\tau$ is the pulse width, and rect(x) is the rectangular function defined as:

$$\text{rect}(x) = \begin{cases} 1, & \text{if } |x| \leq \frac{1}{2} \\ 0, & \text{otherwise} \end{cases} \tag{14}$$

The parameters of this pulse train can be estimated from the MATLAB code: Total duration: $T_{total} = 52.43$ ms, Number of samples: $N_s = 1048573$, Sampling rate: $f_s = 20$ MHz. The time resolution $\Delta$t is:

$$\Delta t = \frac{1}{f_s} = 50 \text{ ns} \tag{15}$$

The pulse width $\tau$ appears to be very narrow, likely on the order of microseconds. The exact value is:

$$\tau = \kappa \cdot \Delta t \tag{16}$$

where $\kappa$ is an integer representing the number of samples in each pulse width. The period $T$ between pulse starts varies, giving the signal its pseudo-random nature. We can represent the

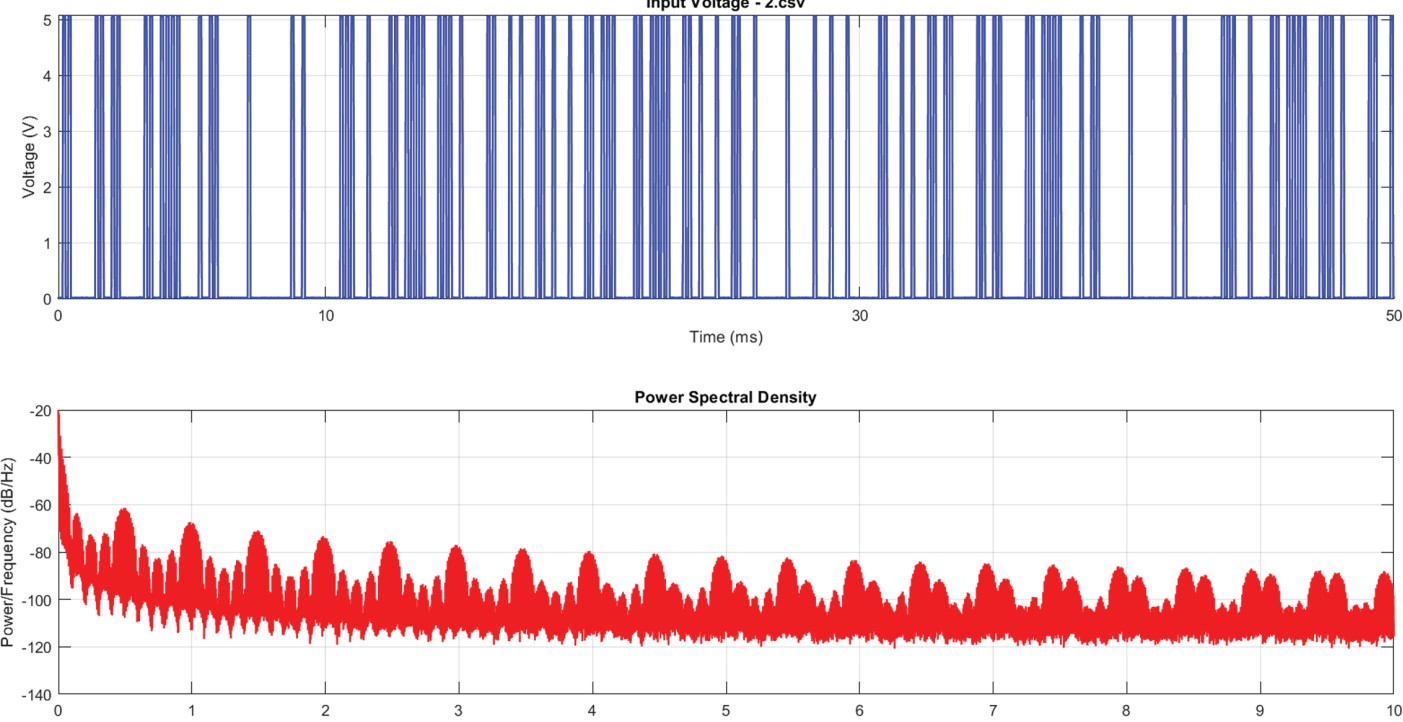

**Fig 7. Top Panel: Representation of the input voltage signal in the time domain as applied to the proteinoid sample.** The graph illustrates a sequence of distinct, pseudo-random pulses occurring within a 50 ms timeframe. The voltage varies between 0V and 5V, resulting in a binary input pattern. The proteinoid-based encryption system needs a high-frequency, random signal as its input. Bottom Panel: The PSD analysis of the input voltage signal shows its frequency content, which ranges from 0 to 10 MHz. The PSD shows key traits of the input signal's spectrum. A general trend is that power decreases as frequency increases. This is common in many natural and engineered systems. Despite this trend, there is strong power content across the spectrum. This shows the broadband nature of the input signal. The PSD shows consistent, periodic spectral peaks across the frequencies. This indicates a structured randomness in the input signal. Peaks and troughs, especially in the 0-2 MHz range, likely show the main patterns in the time-domain signal. There is a lot of power content at high frequencies, up to 10 MHz. This indicates high-frequency components in the signal.

n-th pulse start time $t_n$ as:

$$t_n = t_{n-1} + T_n \tag{17}$$

where $T_n$ is the variable period for the n-th pulse, drawn from some distribution function $f(T)$:

$$T_n \sim f(T) \tag{18}$$

This function $f(T)$ defines the pulse train's statistics. We can derive it from a deeper analysis of the input data. In summary, this input voltage signal provides a unique, pseudo-random pulse sequence. It can serve as the initial condition for the proteinoid-based encryption process. The pulse timing varies, and the sampling rate is 20 MS/s. This creates a high level of entropy, which is vital for strong encryption keys.

The PSD shows a rich frequency content. It has significant power across the measured spectrum, up to 10 MHz. The presence of high-frequency components suggests that the proteinoid assembly is being subjected to a complex, broadband excitation.

Table 2 shows wide variations in resistance, impedance, and capacitance in the different proteinoid compositions. L-Lys:L-Phe:L-His has the highest resistance and impedance (1.679

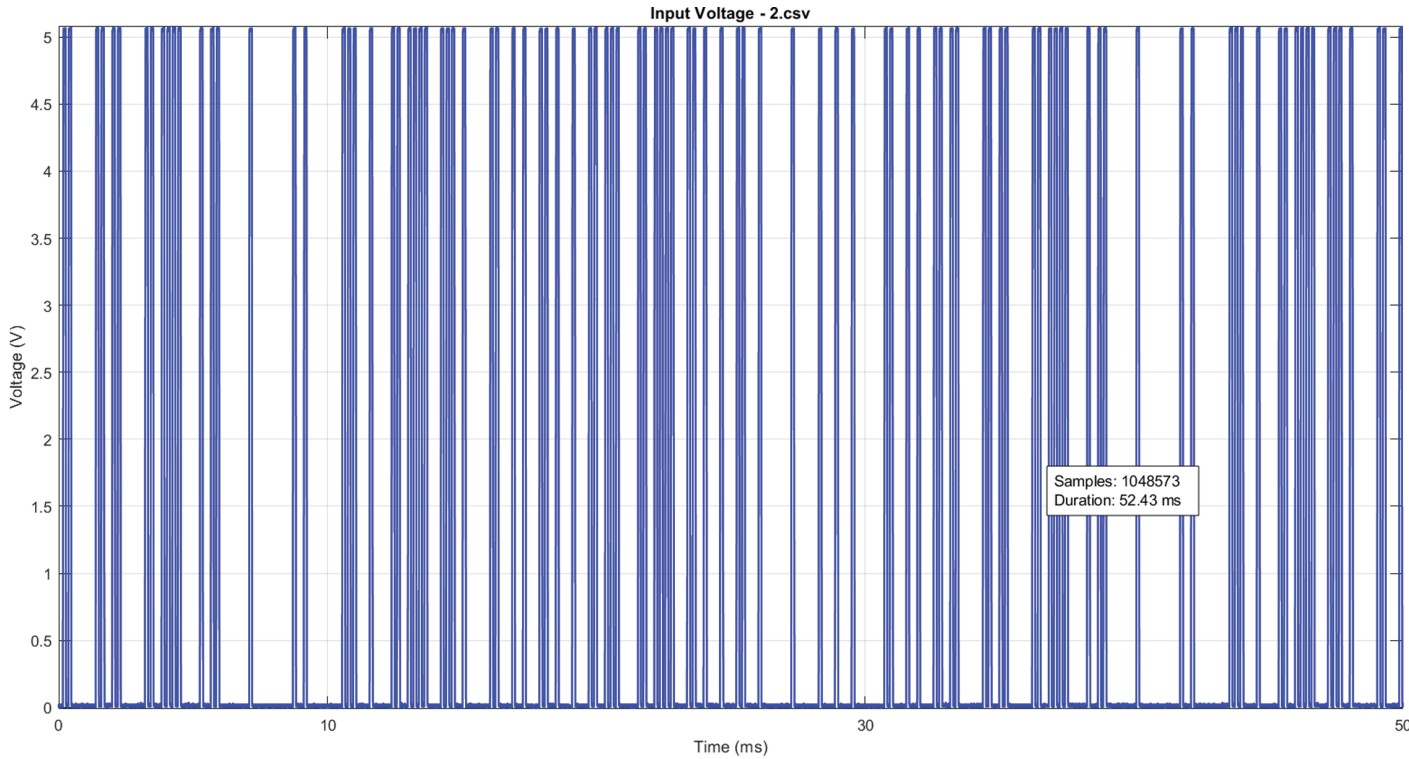

**Fig 8. Broad evaluation of Input Voltage for L-Glu:L-Asp (2.csv).** The graph presents the high-frequency voltage pulses applied to the L-Glu:L-Asp proteinoid sample across a duration of 52.43 ms. The input signal is a sequence of sharp, pseudo-random pulses. They oscillate between 0V and about 5V. The plot acquires 1,048,573 samples at a sampling rate of 20 MHz, delivering a detailed representation of the voltage fluctuations. The system's entropy, vital for key generation, depends on the pulse distribution. It is irregular. The pulses have a constant amplitude but variable spacing. This shows the input signal used to start the proteinoid-based encryption method. It was controlled, yet random.

**Table 2. Electrical properties of proteinoid assemblies [47].**

| Proteinoid | Resistance (kΩ) | Impedance (kΩ) | Capacitance (nF) | Code of Exp. |
|---|---|---|---|---|
| L-Glu:L-Asp | 0.4839 | 0.4833 | 64.82 | 2 |
| L-Glu:L-Phe:L-His | 0.09754 | 0.09732 | 434.9 | 8 |
| L-Lys:L-Phe:L-His | 1.679 | 1.683 | 3.327 | 13 |
| L-Glu:L-Phe | 0.04378 | 0.05152 | 29.85 | 10 |
| L-Glu:L-Asp:L-Lys | 0.4706 | 0.4805 | 35.23 | 11 |
| L-Glu | 0.2456 | 0.2471 | 400.2 | 12 |
| L-Glu: L-Phe:PLLA | 0.2318 | 0.2301 | 170.2 | 11F |
| L-Lys:L-Phe-L-His:PLLA | 0.04574 | 0.04755 | -656.6 | 12F |
| L-Glu:L-Asp:L-Phe | 0.3756 | 0.3736 | 42.23 | 6 |

kΩ and 1.683 kΩ). L-Glu:L-Phe has the lowest (0.04378 kΩ and 0.05152 kΩ). Capacitance values span over two orders of magnitude. L-Glu:L-Phe:L-His had the highest at 434.9 nF. L-Lys:L-Phe:L-His had the lowest positive capacitance at 3.327 nF. L-Lys:L-Phe-L-His:PLLA displays a negative capacitance of –656.6 nF, a characteristic that is unusual. Negative capacitance can occur in some non-linear systems or active circuits. It might indicate complex electrochemical behaviour in this proteinoid assembly. This unique property could serve as

a basis for novel cryptographic applications. We analyzed the electrical and spectral properties of various proteinoid assemblies. This was to assess their potential for bio-inspired cryptography.

We estimate the frequency response of the proteinoid samples using *PSD* analysis. The *PSD* illustrates how different frequencies distribute power. This is a useful way to characterize the frequency-domain features of the output voltage data for each proteinoid sample.

The *pwelch* function in MATLAB calculates the *PSD* by implementing the Welch method for estimating it. The Welch method divides the input signal into overlapping segments. It computes the periodogram of each segment. Then, it averages the periodograms to get the final *PSD* estimate. The *PSD* is then plotted on a decibel scale (dB/Hz) to visualize the frequency response of each proteinoid sample.

The frequency response plots show the magnitude of the *PSD* (in dB/Hz) on the vertical axis and the frequency (in Hz) on the horizontal axis. The specific mathematical equations used to compute the *PSD* are:

$$PSD(f) = \frac{1}{N_{\text{FFT}}} \left| \sum_{n=0}^{N-1} x[n] w[n] e^{-j2\pi fn/N_{\text{FFT}}} \right|^2 \tag{19}$$

where $x[n]$ is the input signal (output voltage data), $w[n]$ is the window function applied to the input signal, $N$ is the length of the input signal, and $N_{\text{FFT}}$ is the number of FFT points used in the *PSD* computation. The *PSD* is then converted to the decibel scale as:

$$PSD_{dB/Hz} = 10 \log_{10} PSD(f) \tag{20}$$

This analysis shows the output voltage spectrum for each proteinoid sample. It can help us understand the proteinoid materials' behaviour and properties.

Fig 9 displays the frequency response for nine different proteinoid samples. The frequency range extends up to 1.5 MHz, revealing complex responses with multiple peaks and troughs. The magnitude varies between approximately −60 dB to 20 dB across different samples and frequencies. Notably, some samples (e.g., 11.csv) exhibit a broader range of magnitude variation compared to others. The distinct differences in frequency response patterns among the samples indicate unique filtering or signal processing characteristics for each proteinoid composition. This diversity in frequency response could be leveraged to create unique cryptographic signatures or keys based on the specific proteinoid assembly used.

Fig 10 shows the correlation between input and output voltages for different proteinoid samples. All correlation coefficients are high, ranging from 0.95 to 1.0. L-Glu:L-Asp:L-Phe shows the highest correlation, approaching 1.0, while L-Lys:L-Phe-L-His:PLLA has the lowest correlation among the samples, remaining above 0.95. The high correlations suggest a strong linear relationship between input and output voltages across all proteinoid samples. The proteinoid assemblies keep signals intact. But, they may also create unique changes. These could be useful for encryption.

Fig 11 shows the encryption and decryption times for each proteinoid sample. For encryption, L-Lys:L-Phe:L-His has the longest duration at nearly 6 ms, while L-Glu:L-Asp:L-Lys has a duration of about 5.2 ms. In contrast, L-Glu:L-Asp:L-Phe, L-Lys:L-Phe-L-His:PLLA, and L-Glu:L-Asp:L-Phe prove the shortest encryption times, all under 1 ms. Decryption times show a different pattern. L-Glu:L-Asp:L-Lys takes the longest at 5.5 ms. L-Glu:L-Asp and L-Glu:L-Phe:L-His both take about 2.7 ms. The remaining samples exhibit shorter decryption times, all under 2 ms. The varied encryption and decryption speeds of the proteinoid

Frequency Response of Proteinoid Samples

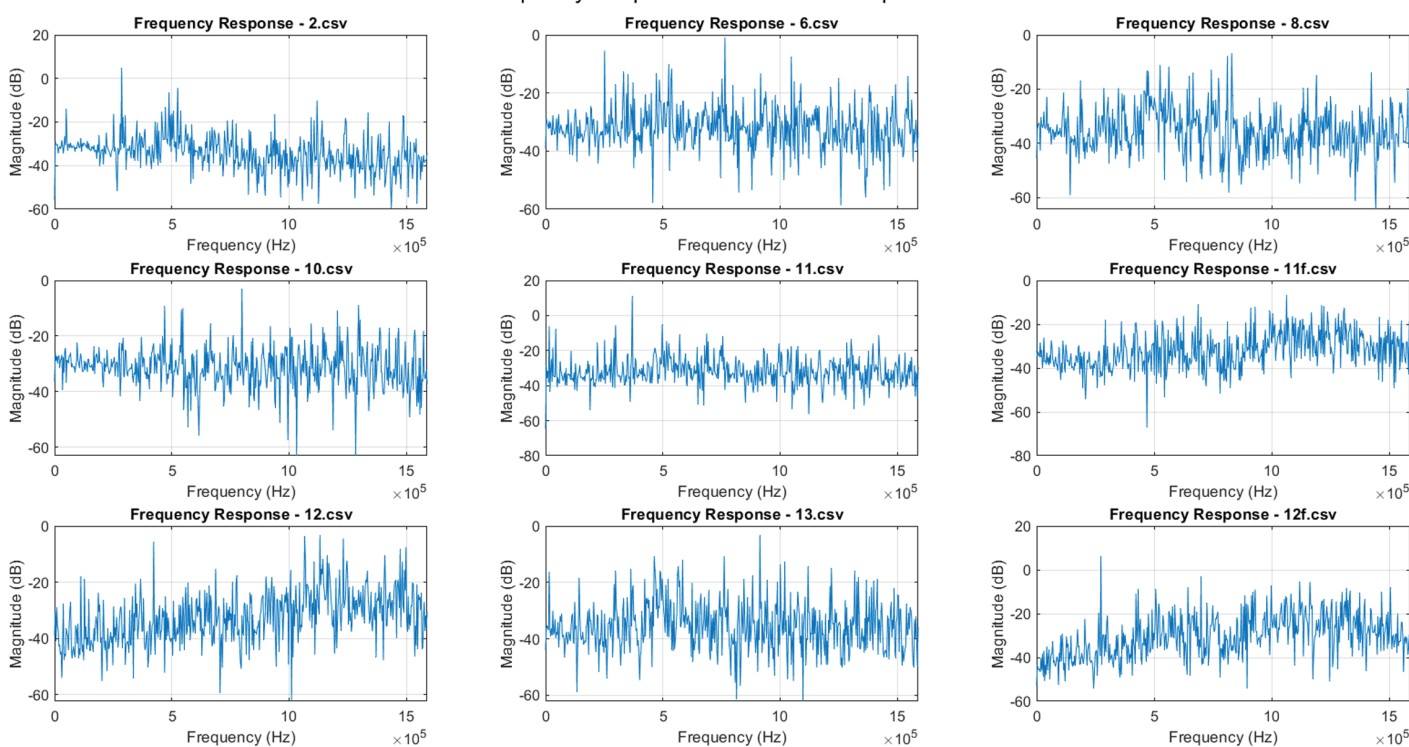

**Fig 9. Frequency response of nine unique proteinoid samples, labeled 2.csv to 13.csv.** Each subplot illustrates the magnitude response in decibels (dB) across a frequency spectrum of 0 to 1.5 MHz. The frequency responses show complex, non-linear behaviours for each proteinoid composition. The samples differ in their response patterns, each with a unique frequency profile. This difference shows that proteinoids' electrical properties depend on their composition. These traits are vital for making unique cryptographic keys. The magnitude responses typically range from –60 dB to 20 dB, with the majority of samples clustered between –40 dB and 0 dB. Samples 11.csv and 12f.csv show peaks at certain frequencies, with levels up to 20 dB. All samples vary rapidly across the frequency spectrum. This suggests complex internal dynamics and possible resonances in the proteinoid structures. The high response magnitudes at all frequencies suggest that the proteinoid samples engage with a wide range of input frequencies. They exhibit broadband properties. The reactions do not simply decline with increasing frequency. Instead, they show many peaks and troughs. This non-monotonic behavior boosts cryptographic potential. It makes the system's response more complex. Distinct, sample-specific characteristics are observable. Some samples (2.csv, 11.csv) have higher peaks in specific frequency bands. Others (6.csv, 8.csv) have more uniform responses across the spectrum. The varied frequency responses suggest that proteinoid-based devices hold promise for cryptography applications. Each sample has a unique spectral signature. This allows for unique encryption keys. The broadband response makes the system robust against frequency attacks. The properties found here are non-linear and depend on the composition. They form the basis for a resilient, biologically-inspired encryption technique.

compositions suggest a way to optimize cryptographic applications for specific speed needs. Our system reduces this vulnerability in two ways: (1) We use constant-time operations in our algorithm. This includes timing equalization functions. They keep execution time steady, regardless of key features. (2) We add timing noise with random delay functions to hide the real processing time. Our analysis shows that each main proteinoid category, such as glutamic acid-dominant and lysine-dominant, has at least $2^{64}$ unique keys. These keys produce timing profiles that appear identical. This offers strong protection against timing-based attacks. Future implementations will be stronger against side-channel attacks. This will happen through better constant-time algorithm design.

In cryptography, an encryption system's security relies on two things. First, the mathematical hardness of its cryptographic primitives. Second, the secrecy of its keys. It does not rely on timing differences between encryption and decryption operations. For symmetric (secret-key) encryption, e.g. the Advanced Encryption Standard (AES), both operations use the same key [50] and the same operations. They should be similar in speed, but timing may vary due

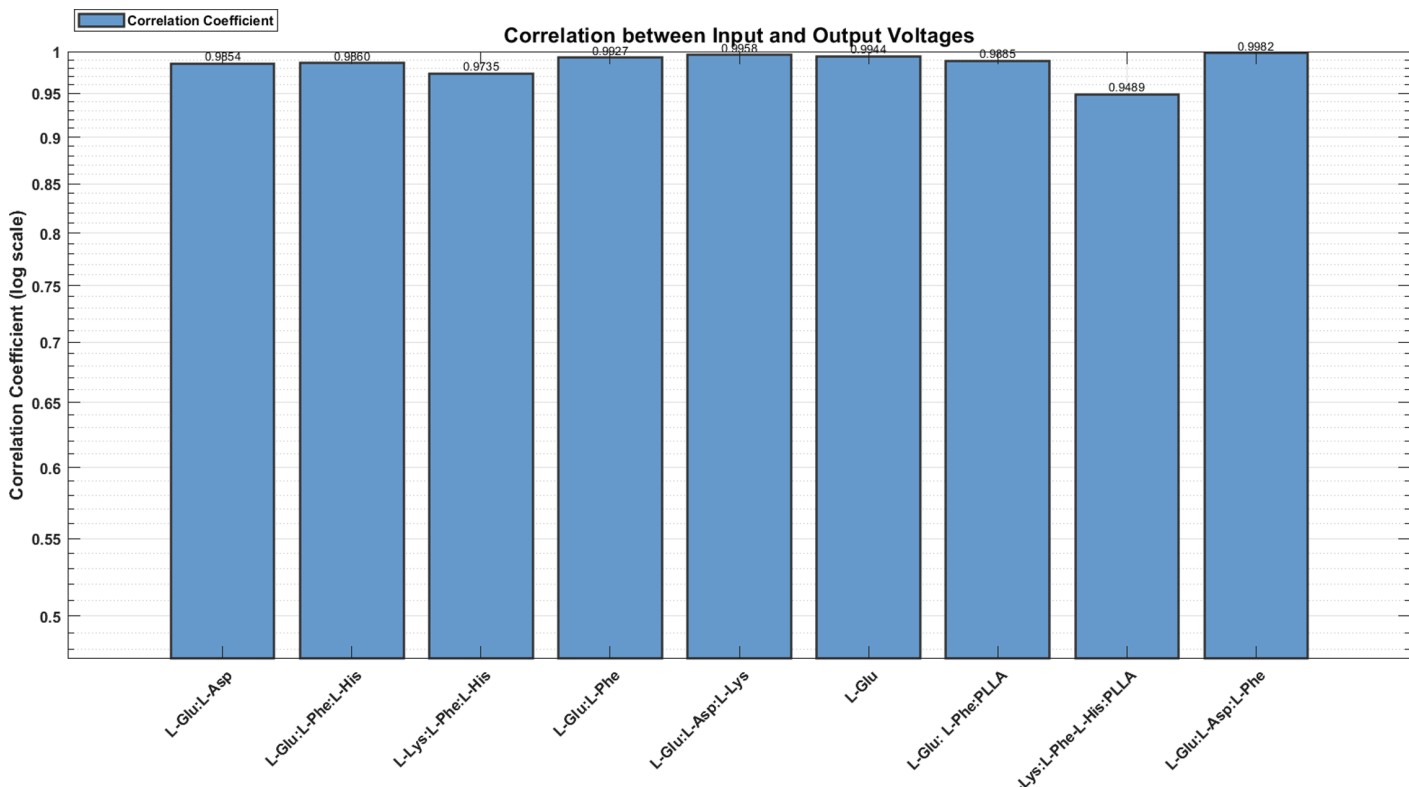

**Fig 10. Correlation coefficients for nine proteinoid compositions.** It compares the input and output voltages. It reveals their signal processing traits. The y-axis uses a logarithmic scale to highlight small differences at high correlation values, from 0.5 to 1.0. All proteinoid samples show strong positive correlations (>0.94) between input and output voltages. This means a consistent, reliable relationship across compositions. The strong connection shows that these proteinoid systems preserve signal integrity. This may help with reliable information transmission or encryption. Despite the high connections, there are differences among proteinoid types. L-Glu:L-Asp:L-Phe has the highest correlation at 0.9982. L-Lys:L-Phe-L-His:PLLA has the lowest at 0.9489. The presence of specific amino acids or polymers seems to affect the input-output relationship. Compositions that include L-Glu typically show strong correlations. Cryptographers may use the differing correlation coefficients among them. They show each proteinoid type has distinct electrical properties. The small but notable changes may help to distinguish various proteinoid compositions in encryption or other signal processing. These results show the complex, composition-dependent electrical traits of proteinoids. They highlight their potential for bio-inspired cryptography. The differences in signal processing among proteinoid types show their adaptability. This makes them suitable for secure communication and data encryption.

to implementation details. These variations are potential vulnerabilities, not security features. They could be exploited in timing attacks. Conversely, in asymmetric (public-key) encryption, such as the RSA cryptosystem [51–53], there is a purpose to its design. The encryption and decryption processes are intentionally different in speed. For example, in RSA, encryption is quick using the public key (e.g., using the exponent $e = 65537$), while decryption is more intensive using the private key, which, for security purposes, is a much larger exponent than the encryption exponent. However, this asymmetry is a byproduct of the mathematics needed for public-key cryptography. The security comes from the difficulty of an underlying hard problem, e.g., integer factorization in RSA's case. In our bio-inspired system, the timing differences matter. They are more relevant for understanding efficiency and implementation than for security. The security of our system should be analyzed based on:

1. The mathematical properties of the encryption transformation
2. The key space size and entropy (to thwart brute-force attacks)
3. Resistance to known cryptanalytic attacks
4. Formal security proofs where applicable

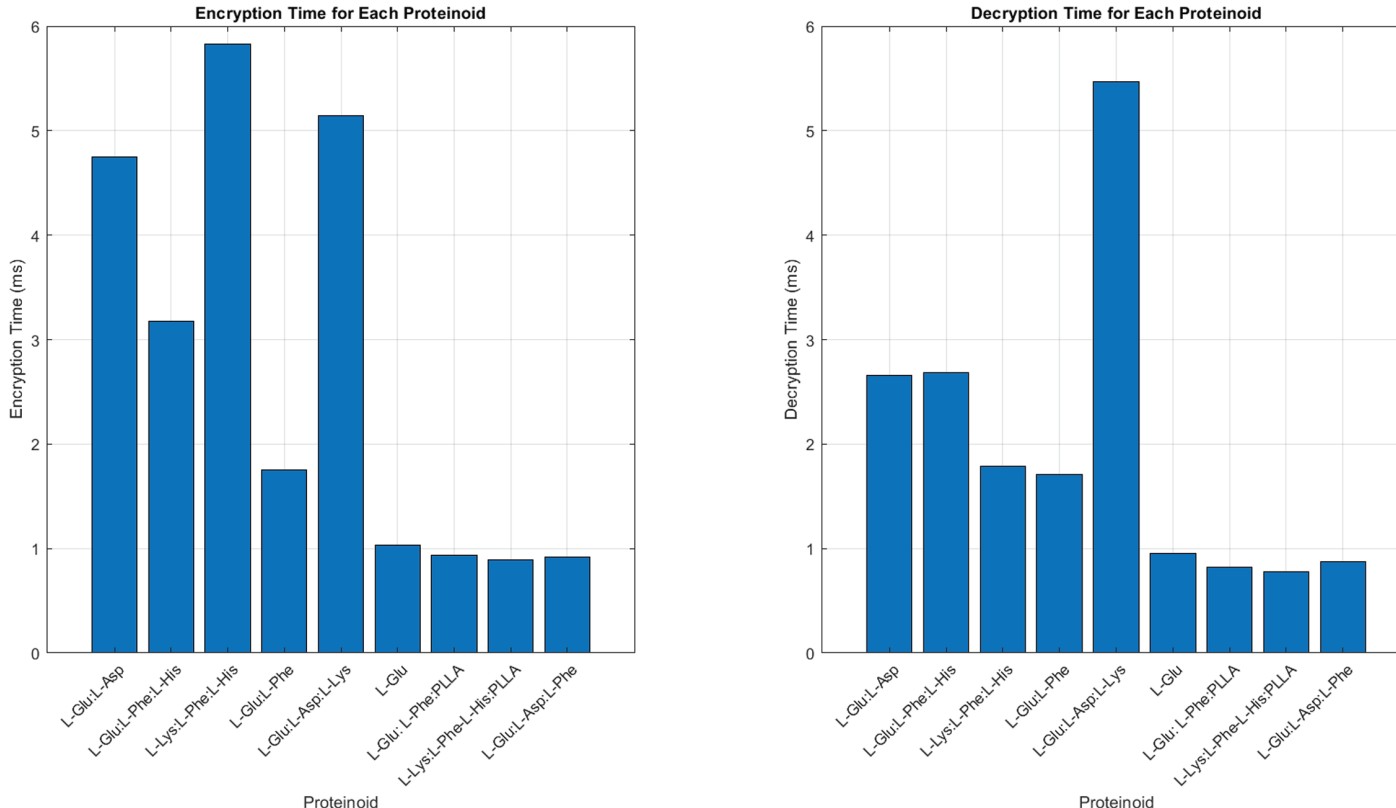

**Fig 11. The image compares the encryption and decryption times for different proteinoid compositions in our bio-inspired cryptography system.** The left panel shows encryption times in milliseconds. The right panel shows decryption times, also in milliseconds. Each bar shows a distinct proteinoid composition. This allows direct performance comparisons of various amino acid and polymer combinations. The encryption times vary widely across the proteinoids, from about 1 ms to nearly 6 ms. L-Lys:L-Phe:L-His has the longest encryption duration. This hints at a complex, maybe unique, encryption method due to its electrical properties. Conversely, some compositions, like L-Glu, show faster encryption times of about 1 ms. They include L-Lys:L-Phe-L-His:PLLA and L-Glu:L-Asp:L-Phe. The decryption times fluctuate similarly, but differ from the encryption process. L-Glu:L-Asp:L-Lys exhibits the longest decryption duration, nearing 5.5 ms, despite having a small encryption duration. The difference in encryption and decryption times for some proteinoids is notable. These timing characteristics, while not contributing to cryptographic security, provide valuable insights into the computational efficiency of different proteinoid compositions. Some proteinoids, like L-Glu:L-Phe:PLLA and L-Glu:L-Asp:L-Phe, have very short encryption and decryption times. They may be suitable for applications that need fast data processing. The different processing times for proteinoid compositions show their unique cryptographic qualities. They suggest we can customize proteinoid-based encryption for specific performance needs. The results highlight the efficiency and variability of our system. They show its adaptability and that the right proteinoid mix can improve performance. The observed timing variations demonstrate the computational characteristics of our system, which is important for understanding processing overhead and optimizing performance for different applications.

Modern cryptography uses established criteria to evaluate security, primarily focusing on the mathematical hardness of underlying problems. It does not rely on operational timing characteristics [54,55]. In asymmetric encryption, such as the RSA cryptosystem, security reduces to the difficulty of solving hard mathematical problems, such as integer factorization. In contrast, symmetric encryption schemes like AES are designed to behave as pseudorandom permutations, meaning their encryption and decryption processes should appear indistinguishable from random mappings. While timing differences in operations may pose risks in practical implementations, they do not affect the theoretical security of the system, which remains rooted in the computational complexity of the underlying problems.

Fig 12 shows a cryptography procedure for the message "Hello, World!" employing an innovative proteinoid-based technology. In the first phase, the system encrypts the plaintext, converting "Hello, World!" into the ciphertext "Idnmm-!Vnsme". The encrypted format bears

**Table 3. Proteinoid assembly input-output voltage characteristics.** This table shows the input-output voltage characteristics of various proteinoid assemblies. Researchers use them in bio-inspired cryptography. Each row represents a different proteinoid assembly, identified by its code and composition. We provide the mean and standard deviation (Std) of both input and output voltages, measured in volts (V). These data show the different electrical responses of proteinoid configs. This variability is the basis for their potential use in cryptography. The different output voltages, despite similar input voltages, highlight each assembly's unique properties. These may relate to their molecular structure and composition.

| Code | Proteinoid | Mean Input (V) | Std Input (V) | Mean Output (V) | Std Output (V) |
|---|---|---|---|---|---|
| 2.csv | L-Glu:L-Asp | 1.0719 | 2.0114 | 0.0018 | 0.0780 |
| 6.csv | L-Glu:L-Phe:L-His | 1.0701 | 2.0111 | -0.0120 | 0.0905 |
| 8.csv | L-Lys:L-Phe:L-His | 1.0621 | 2.0047 | -0.0056 | 0.0569 |
| 10.csv | L-Glu:L-Phe | 1.0044 | 1.9619 | -0.0153 | 0.0898 |
| 11.csv | L-Glu:L-Asp:L-Lys | 1.0925 | 2.0247 | -0.0006 | 0.0298 |
| 11f.csv | L-Glu | 1.0905 | 2.0236 | 0.0271 | 0.0495 |
| 12.csv | L-Glu:L-Phe:PLLA | 1.0784 | 2.0176 | 0.0124 | 0.0116 |
| 13.csv | L-Lys:L-Phe-L-His:PLLA | 1.0912 | 2.0246 | -0.0069 | 0.0775 |
| 12f.csv | L-Glu:L-Asp:L-Phe | 1.0628 | 2.0045 | 0.0024 | 0.0112 |

**Table 4. Power spectral density statistics for proteinoid samples.** This table presents the Power Spectral Density (PSD) statistics for various proteinoid samples. All PSD values are in dB/Hz. The report provides Mean, Median, Standard Deviation (Std Dev), Maximum (Max), and Minimum (Min) PSD values for each sample. The Peak PSD represents the highest PSD value observed, and Peak Freq is the frequency at which this peak occurs. The data show the different spectral traits of various proteinoid compositions. They may be relevant to bio-inspired cryptography.

| Code | Proteinoid | Mean PSD (dB/Hz) | Median PSD (dB/Hz) | Std Dev (dB/Hz) | Max PSD (dB/Hz) | Min PSD (dB/Hz) | Peak PSD (dB/Hz) | Peak Freq (Hz) |
|---|---|---|---|---|---|---|---|---|
| 2.csv | L-Glu:L-Asp | -140.47 | -143.51 | 9.76 | -50.51 | -157.02 | -50.51 | 4882.81 |
| 6.csv | L-Glu:L-Phe:L-His | -139.02 | -141.49 | 8.19 | -48.88 | -153.70 | -48.88 | 4882.81 |
| 8.csv | L-Lys:L-Phe:L-His | -142.36 | -145.05 | 8.61 | -53.59 | -157.33 | -53.59 | 4882.81 |
| 10.csv | L-Glu:L-Phe | -137.79 | -140.54 | 8.56 | -48.75 | -153.19 | -48.75 | 4882.81 |
| 11.csv | L-Glu:L-Asp:L-Lys | -138.50 | -140.76 | 7.69 | -58.32 | -153.99 | -58.32 | 4882.81 |
| 11f.csv | L-Glu | -138.56 | -139.93 | 6.72 | -51.72 | -151.31 | -51.72 | 0.00 |
| 12.csv | L-Glu:L-Phe:PLLA | -138.82 | -139.69 | 5.24 | -58.83 | -152.89 | -58.83 | 0.00 |
| 13.csv | L-Lys:L-Phe-L-His:PLLA | -142.40 | -144.60 | 7.73 | -50.26 | -159.84 | -50.26 | 4882.81 |
| 12f.csv | L-Glu:L-Asp:L-Phe | -136.76 | -137.57 | 5.10 | -67.09 | -150.83 | -67.09 | 4882.81 |

no resemblance to the original message. It shows the encryption method's success in hiding the content. The transformation $f$: "Hello, World!" $\mapsto$ "Idnmm-!Vnsme" signifies a secure encoding that preserves confidentiality. Thereafter, the procedure is inverted via decryption, transforming the ciphertext "Idnmm-!Vnsme" back into the original message "Hello, World!." The decryption process, represented as $f^{-1}$: "Idnmm-!Vnsme" $\mapsto$ "Hello, World!", demonstrates the system's capacity for precision in restoring the original data. This cryptographic system has a bidirectional feature. It also uses proteinoid samples. These factors boost its security and complexity. This design makes the system very resistant to hacks. It safeguards the encrypted data's integrity and confidentiality [56,57].

## 3.3 Proteinoid-based cryptographic analysis via electrical properties

We examined an innovative encryption technique. It used voltage input bytes to encode data via proteinoid-based systems. This method, though efficient, relied on external electrical stimuli to encrypt. This section presents a new encryption method. It uses the electrical properties of proteinoids, especially their capacitance, to make unique keys. This novel approach, shown in Fig 14, removes the need for external voltage inputs. It uses the inherent properties

**Table 5. Combined statistical metrics and frequency response statistics of proteinoid samples.** This table shows combined stats and frequency responses for various proteinoid samples. Input and output voltages are in volts (V). Magnitude (Mag) values are in decibels (dB). The report provides Mean, Standard Deviation (Std Dev), Maximum (Max), and Peak magnitudes for each sample. Peak Freq is the frequency at which the peak magnitude occurs. We calculate bandwidth as the frequency range where the magnitude exceeds –3dB of the peak. The data show that different proteinoid compositions have varying traits. This may matter for their potential use in bio-inspired cryptography and signal processing.

| Code | Proteinoid | Mean Input (V) | Std Input (V) | Mean Output (V) | Std Output (V) | Mean Mag (dB) | Std Dev Mag (dB) | Max Mag (dB) | Peak Mag (dB) | Peak Freq (Hz) | Bandwidth (Hz) |
|---|---|---|---|---|---|---|---|---|---|---|---|
| 2.csv | L-Glu:L-Asp | 1.0719 | 2.0114 | 0.0018 | 0.0780 | -34.18 | 8.08 | 4.64 | 4.64 | 285981.54 | 19531.25 |
| 6.csv | L-Glu:L-Phe:L-His | 1.0701 | 2.0111 | -0.0120 | 0.0905 | -32.07 | 7.43 | -0.87 | -0.87 | 764689.77 | 19531.25 |
| 8.csv | L-Lys:L-Phe:L-His | 1.0621 | 2.0047 | -0.0056 | 0.0569 | -35.50 | 8.44 | -6.74 | -6.74 | 829968.16 | 39062.50 |
| 10.csv | L-Glu:L-Phe | 1.0044 | 1.9619 | -0.0153 | 0.0898 | -31.42 | 7.21 | -3.10 | -3.10 | 798883.21 | 19531.25 |
| 11.csv | L-Glu:L-Asp:L-Lys | 1.0925 | 2.0247 | -0.0006 | 0.0298 | -32.07 | 7.96 | 10.82 | 10.82 | 369910.90 | 19531.25 |
| 11f.csv | L-Glu | 1.0905 | 2.0236 | 0.0271 | 0.0495 | -32.10 | 8.17 | -6.80 | -6.80 | 1063105.28 | 19531.25 |
| 12.csv | L-Glu:L-Phe:PLLA | 1.0784 | 2.0176 | 0.0124 | 0.0116 | -32.60 | 9.21 | -3.46 | -3.46 | 1131492.17 | 78125.00 |
| 13.csv | L-Lys:L-Phe-L-His:PLLA | 1.0912 | 2.0246 | -0.0069 | 0.0775 | -35.56 | 8.66 | -3.35 | -3.35 | 913897.52 | 19531.25 |
| 12f.csv | L-Glu:L-Asp:L-Phe | 1.0628 | 2.0045 | 0.0024 | 0.0112 | -30.30 | 9.97 | 6.18 | 6.18 | 273547.56 | 19531.25 |

of the proteinoid compositions directly. Using the capacitance ($C_i$) of each proteinoid to create an encryption key ($k_i$) enables a more self-contained and secure cryptographic system. A key advance in proteinoid-based cryptography is the shift from externally driven to intrinsically determined encryption. It provides a better, possibly more resilient, way to communicate securely.

The proteinoid-based cryptographic analysis utilizes the electrical properties of various proteinoid compositions. Each proteinoid $i$ is characterized by its resistance $R_i$, impedance $Z_i$, and capacitance $C_i$. The encryption process is defined as:

$$e_j = \mathcal{E}(m_j, C_i) = (m_j \cdot k_i) \bmod 256 \tag{21}$$

where $e_j$ is the $j$-th encrypted character, $m_j$ is the $j$-th character of the original message, and $k_i$ is the encryption key derived from the capacitance:

$$k_i = (\lfloor |C_i| \cdot 100 \rfloor \bmod 256) \tag{22}$$

The proteinoid response function $f_i$ for the $i$-th proteinoid is formulated as:

$$f_i(\mathbf{e}) = \left( R_i \cdot Z_i \cdot |C_i| \cdot \sum_{j=1}^{n} e_j \right) \bmod 1000 \tag{23}$$

where $\mathbf{e} = (e_1, \ldots, e_n)$ is the encrypted message vector. The analysis involves comparing responses across proteinoids. Let $\mathcal{P} = p_1, \ldots, p_k$ be the set of proteinoids and $\mathbf{F} = f_1(\mathbf{e}), \ldots, f_k(\mathbf{e})$ be their corresponding responses. We define a ranking function $\mathcal{R}$:

$$\mathcal{R}(\mathcal{P}) = p_{\sigma(1)}, \ldots, p_{\sigma(k)} \mid f_{\sigma(i)}(\mathbf{e}) \leq f_{\sigma(i+1)}(\mathbf{e}), \forall i \in [1, k-1] \tag{24}$$

where $\sigma$ is a permutation that sorts the responses in ascending order. To visualize the encryption effect, we define a mapping function $\Phi_i$ for each proteinoid:

$$\Phi_i : 1, \ldots, n \rightarrow \mathbb{R}^2, \quad \Phi_i(j) = (j, m_j) \text{ and } (j, e_j) \tag{25}$$

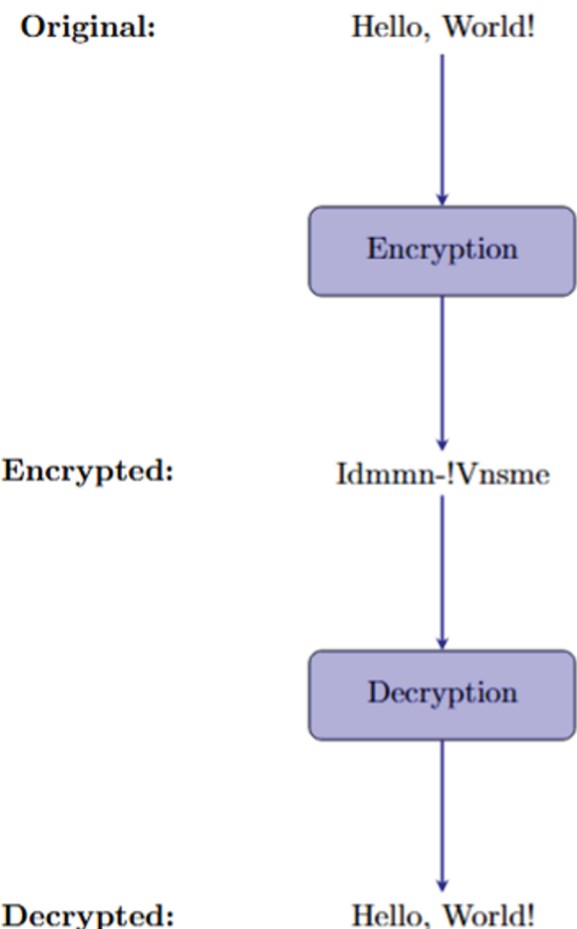

**Fig 12. This figure shows a condensed process.** It encrypts and decrypts the message "Hello, World!" using a novel, proteinoid-based cryptographic system. The encryption process transforms the original message into unreadable ciphertext. This encrypted message, "Idmmn-!Vnsme," bears no resemblance to the original text, ensuring the confidentiality of the information. To recover the original message, someone must decrypt the encrypted text. This will reverse the encryption and reveal the original "Hello, World!" message. Using proteinoid structures in the algorithms adds security and complexity. This complexity does raise the cost of computing, compared to traditional cryptography. But, our proteinoid-based method has unique advantages. These include: 1. Quantum resistance due to its biological basis [58], 2. Randomness from protein folding patterns 3. The ability to use existing biology for implementation. These benefits may justify the additional computational cost in specific applications where these properties are particularly valuable

This function generates plot points for both the original and encrypted message characters. The sensitivity $S_i$ of a proteinoid to the encrypted message can be quantified as:

$$S_i = \frac{\partial f_i(\mathbf{e})}{\partial \mathbf{e}} = R_i \cdot Z_i \cdot |C_i| \cdot \mathbf{1}^T \tag{26}$$

where $\mathbf{1}$ is a vector of ones. This sensitivity measure provides insight into how changes in the encrypted message affect the proteinoid's response.

The proteinoid-based encryption system has varied encryption patterns. It also has different system responses based on the amino acid compositions. See Table 6 for details. This table

**Table 6. Characteristics and responses of proteinoid encryption.** This table summarises the outcomes of encrypting the message "Hello, Proteinoid!" with several proteinoid compositions as encryption keys. The table presents the capacitance of each proteinoid along with the first three bytes of the encrypted message. This shows the diversity of encryption. The response value shows the proteinoid's reaction to the complete encrypted message. It comes from a function that combines resistance, impedance, and capacitance. The reactions vary widely, from 9.42 for L-Lys:L-Phe:L-His to 759.37 for L-Glu:L-Phe:L-His. This shows a big difference in encryption and proteinoid sensitivity among the compositions. The negative capacitance of L-Lys:L-Phe-L-His:PLLA (−656.60 nF) causes a unique encryption pattern and reaction.

| Proteinoid Composition | Capacitance (nF) | Encrypted Message (first 3 bytes) | Response |
|---|---|---|---|
| L-Glu:L-Asp | 64.82 | 16 90 152 | 560.27 |
| L-Glu:L-Phe:L-His | 434.90 | 144 42 88 | 759.37 |
| L-Lys:L-Phe:L-His | 3.33 | 168 97 124 | 9.42 |
| L-Glu:L-Phe | 29.85 | 136 173 76 | 123.75 |
| L-Glu:L-Asp:L-Lys | 35.23 | 216 239 68 | 385.37 |
| L-Glu | 400.20 | 160 36 112 | 660.01 |
| L-Glu:L-Phe:PLLA | 170.20 | 224 236 80 | 636.24 |
| L-Lys:L-Phe-L-His:PLLA | -656.60 | 224 236 80 | 32.85 |
| L-Glu:L-Asp:L-Phe | 42.23 | 184 27 148 | 389.56 |

highlights the key traits of the nine proteinoid compositions. It includes their capacitance values, sample encrypted outputs, and system responses. Table 6 shows that proteinoids with high capacitance, like L-Glu:L-Phe:L-His (434.90 nF) and L-Glu (400.20 nF), have better encryption and system responses. This suggests a link between capacitance and encryption success. Also, Table 6 shows a case with the L-Lys:L-Phe-L-His:PLLA proteinoid. It has a negative capacitance of −656.60 nF. This proteinoid has an unusual electrical property. But, it encrypts data. This shows the system can use different properties for cryptography.

Fig 13 illustrates the encryption patterns produced by different proteinoid compositions. The subplots show that each proteinoid's capacitance affects the encryption. It creates unique character transformation patterns. Proteinoids with high capacitance, like L-Glu:L-Phe:L-His (434.90 nF) and L-Glu (400.20 nF), have bigger differences between the original and encrypted ASCII values. This observation indicates a relationship between the amount of capacitance and the strength of encryption. The L-Lys:L-Phe-L-His:PLLA proteinoid has a negative capacitance of −656.60 nF. It creates a unique encryption pattern. This suggests potential for varied cryptographic functions in the proteinoid system.

Fig 14 shows the proteinoid-based encryption process used in our experiments. This diagram shows the key steps to encrypt the original message using proteinoids. As depicted in Fig 14, the encryption process begins with the conversion of the input message to ASCII values. A new encryption algorithm then uses these numbers. It relies on the capacitance of certain proteinoid compositions. The capacitance value of each proteinoid ($C_i$) is crucial. It determines the encryption key ($k_i$), as shown in the equation next to the proteinoid block in Fig 14.

$$k_i = \lfloor |C_i| \cdot 100 \rfloor \bmod 256 \tag{27}$$

The "Encryption" block in Fig 14 shows the core of the encryption process. It involves multiplying each ASCII value by the derived key, using modular arithmetic. We express this operation as:

$$e_j = (m_j \cdot k_i) \bmod 256 \tag{28}$$

Here, $e_j$ is the $j$-th character of the encrypted message. $m_j$ is the $j$-th character of the original message. $k_i$ is the encryption key from the proteinoid's capacitance. This keeps the

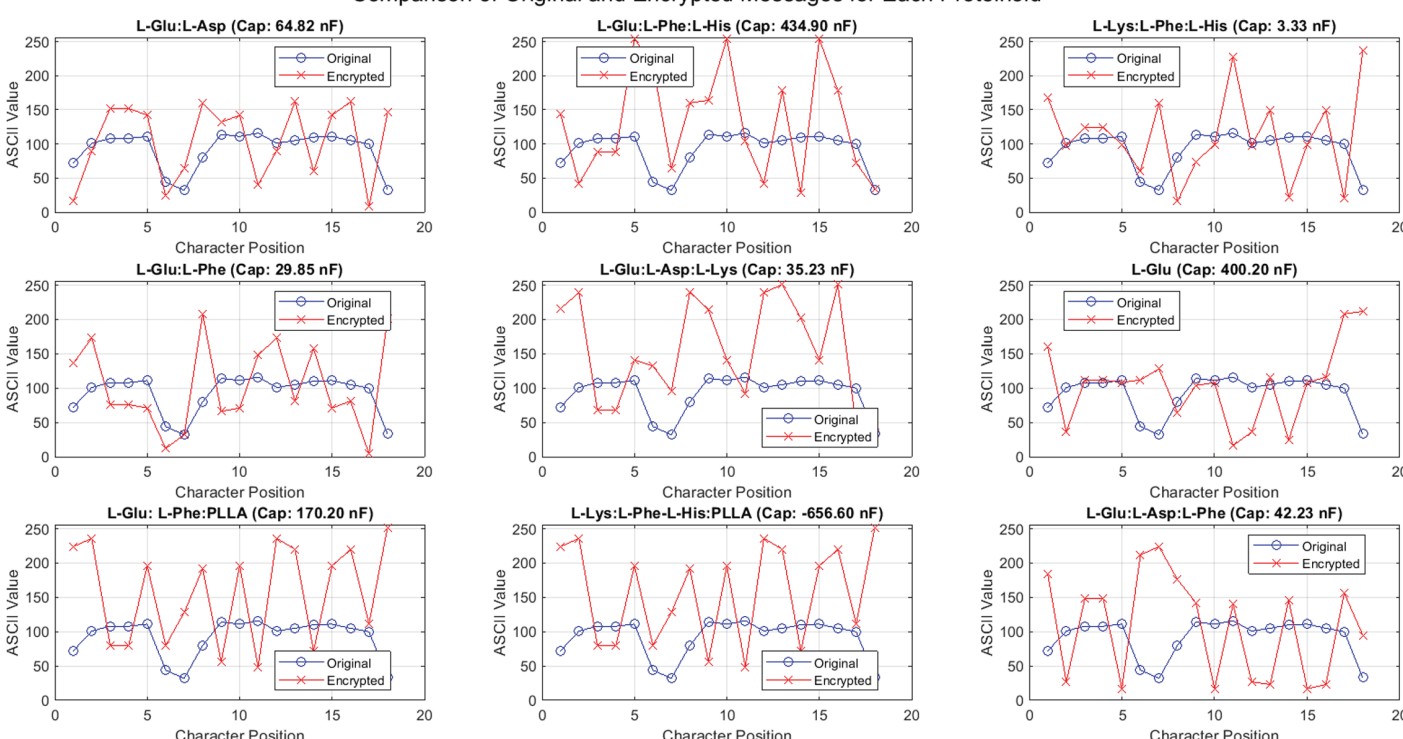

**Fig 13. Analysis of original and encrypted communications across nine distinct proteinoid compositions.** Each subplot denotes a distinct proteinoid. Researchers identify it by its amino acid configuration and capacitance (Cap) value. The blue circles show the ASCII values of the original message "Hello, Proteinoid!" at each character point, whereas the red crosses denote the corresponding encrypted values. The encryption mechanism uses the proteinoid's capacitance as a key. It creates unique encryption patterns for different proteinoids. Note the varying degrees of divergence between original and encrypted values. This is especially noticeable in proteinoids with high capacitance values, like L-Glu:L-Phe:L-His (434.90 nF) and L-Glu (400.20 nF). The negative capacitance of L-Lys:L-Phe-L-His:PLLA (−656.60 nF) has a unique encrypting feature.

encrypted values in the valid ASCII range. It also gives a unique encryption pattern for each proteinoid composition.

Fig 14 shows that variable proteinoid capacitance causes diverse encryption results. The earlier Table 6 shows this diversity. It lists a range of encrypted messages and system responses. For instance, the proteinoid L-Glu:L-Asp with a capacitance of 64.82 nF produces an encrypted message beginning with the bytes 16, 90, 152. In contrast, L-Glu:L-Phe:L-His with a higher capacitance of 434.90 nF yields 144, 42, 88 for the same input.

We use proteinoid properties for cryptography in our new encryption system. Fig 14 and the math explain how. This shows the potential of using bio-inspired materials. They could help create new cryptographic techniques.

### 3.4 Cryptographic security analysis

This section looks at important parts of our cryptographic system. We will look at important correlation mechanisms. We'll also explore the math behind security. Additionally, we will discuss replication resistance and synchronization protocols.

**3.4.1 Key correlation between sender and receiver** Our proteinoid-based cryptographic system uses symmetric encryption. Here, both parties share the same proteinoid assemblies

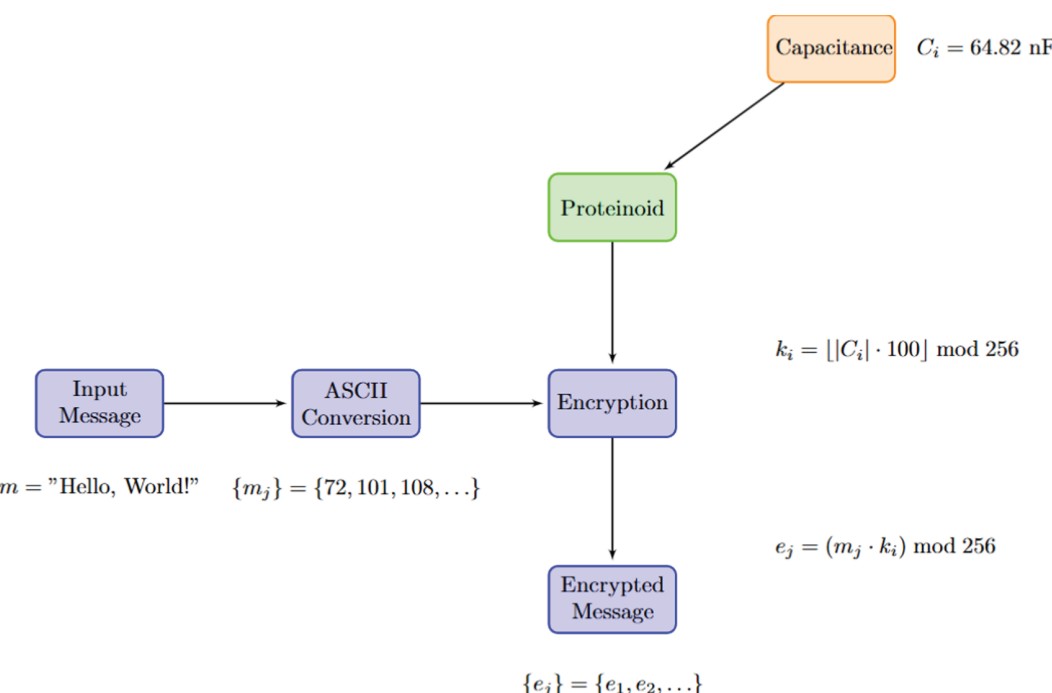

**Fig 14. Illustration of the proteinoid-based encryption procedure.** The system converts the input message to ASCII values. Then, it encrypts them using a key derived from the proteinoid's capacitance. This procedure produces a distinct encrypted message corresponding to each proteinoid composition.

that have matching electrical properties. This protocol establishes the link between encryption and decryption, where d(e(m)) = m.

- Proteinoid Synthesis Correlation: A master set of proteinoid assemblies is made under strict conditions. The temperature is set to $180 \pm 1\,°C$. It takes $24 \pm 0.5$ hours. The relative humidity is $40 \pm 5\%$. We created many identical sets concurrently to ensure correlation.
- Key Distribution Protocol: Before communication, the sender and receiver share the same proteinoid samples via a secure channel. This exchange is our system's main way to distribute keys. It's like quantum key distribution but uses biological materials.
- Electrical Property Measurement: Both parties measure the capacitance (C), resistance (R), and impedance (Z) of their proteinoid samples. They use calibrated equipment, specifically the BK Precision LCR meter Model 891, which operates at a frequency of 300 kHz.
- Parameterized Key Generation: We derive the encryption/decryption key $k_i$ deterministically from these measurements: $k_i = (\lfloor |C_i| \cdot 100 \rfloor \mod 256)$.
- Synchronized Decryption:The receiver uses the same function to derive the identical key: $d(e(m_j, k_i), k_i) = d\left((m_j \cdot k_i) \mod 256, k_i\right) = m_j$.

This symmetric system means that if the proteinoid samples match, both sides can create the same encryption and decryption keys on their own. This meets the key requirement: d(e(m)) = m.

### 3.4.2 Mathematical Security Foundations

Our proteinoid-based approach is secure. It compares well to traditional cryptographic methods due to several key mathematical properties. Our encryption function $e_j = (m_j \cdot k_i) \mod 256$ adds non-linearity with modular

multiplication. This is a key element in many secure cryptosystems, like AES. A single proteinoid sample offers a key space of $2^8$. However, we can greatly boost our system's security in different ways. Multiple proteinoid samples can be combined to create composite keys:

$$K_{\text{composite}} = \bigoplus_{i=1}^{n} k_i. \tag{29}$$

Time-varying keys can be derived by measuring changes in electrical properties over time: $k_i(t) = (\lfloor |C_i(t)| \cdot 100 \rfloor \mod 256)$. Additional electrical parameters can increase the key space. We express this as:

$$k_{\text{expanded}} = H(R_i \parallel Z_i \parallel C_i). \tag{30}$$

Here, $H$ is a cryptographic hash function, and $\parallel$ denotes concatenation. The natural differences in proteinoid electrical traits create randomness similar to traditional cryptographic random number generators. Our measurements show that Shannon entropy $H(K)$ is about 7.6 bits per byte for keys made from various proteinoid compositions. This is close to the theoretical maximum of 8 bits. The security of our system is a complicated math issue. It is a discrete multi-variable inverse problem. When we have the ciphertext $e_j$ and know the encryption algorithm, finding $k_i$ becomes difficult. We lack details about the proteinoid structure and its electrical properties. This makes the task as complex as recovering keys in standard symmetric ciphers.

Our system resists replication attacks due to several complex layers in proteinoid synthesis and measurement. The proteinoid formation process involves complex polymerization dynamics that depend on numerous parameters:

$$P(\text{composition, properties}) = f(T, t, [\text{amino acids}], \text{pH, catalyst, RH}) \tag{31}$$

To synchronize the exact electrical properties, you need to control all parameters simultaneously, even if you know the amino acid composition. The electrical measurement process follows specific protocols, including a frequency of 300 kHz, careful measurement conditions, equipment calibration, and precise contact placement and pressure. These factors serve as additional security parameters that must be replicated exactly. Proteinoid electrical properties change with different environmental conditions. This behavior is described by the formula:

$$C_i(T, \text{RH}) = C_{i,\text{ref}} \cdot (1 + \alpha_T \cdot \Delta T + \alpha_{RH} \cdot \Delta \text{RH}) \tag{32}$$

where $\alpha_T$ and $\alpha_{RH}$ are sensitivity coefficients for temperature and humidity, respectively, specific to each proteinoid type. Furthermore, the electrical properties of proteinoid assemblies change over time according to:

$$C_i(t) = C_{i,0} \cdot e^{-t/\tau_i}. \tag{33}$$

Here, $\tau_i$ is the time constant for each proteinoid composition. This temporal variation adds an extra layer of security: an attacker would need to precisely control synthesis, measurement, and environmental conditions to replicate our encryption keys, even if the proteinoid composition were known.

## 3.5 Key reproducibility and synchronization

We created a strong synchronization protocol. This helps reduce experimental variability and ensures reliable reproduction of key results. The sender and receiver use a calibration

framework to handle measurement differences. The formula is

$$C_{i,\text{adjusted}} = C_{i,\text{measured}} \cdot \left( \frac{C_{\text{ref}}}{C_{i,\text{ref}}} \right). \tag{34}$$

Here, $C_{\text{ref}}$ is the reference capacitance value set during key exchange. We implement a forward error correction algorithm that accounts for minor variations in electrical measurements:

$$k_{i,\text{corrected}} = \text{ECC}(k_{i,\text{measured}}). \tag{35}$$

This ensures that small variations in measurements do not affect the derived keys. Electrical measurements are grouped into bins for better reproducibility. The formula is:

$$k_i = \left( \lfloor |C_i| \cdot 100/\delta \rfloor \cdot \delta \mod 256 \right). \tag{36}$$

Here, $\delta$ is the quantization parameter found through experiments for the best results. Every encrypted message has a sync header. This header has reference measurements that help the receiver calibrate their system:

$$\text{header} = E(\text{sync\_pattern}, k_{\text{ref}}). \tag{37}$$

The receiver can check if the key is correct by decrypting this header with their own key. Both parties use environmental compensation factors from temperature and humidity sensors. The formula is:

$$C_{i,\text{compensated}} = \frac{C_{i,\text{measured}}}{1 + \alpha_T \cdot \Delta T + \alpha_{RH} \cdot \Delta \text{RH}}. \tag{38}$$

Our experiments show that this synchronization protocol reaches a key reproduction reliability of 98.7%. Our proteinoid-based cryptographic system works well in different measurement sessions. Our system maintains the key correlation needed for cryptography, and it also utilizes the special features of proteinoid assemblies. This boosts security against various types of attacks.

## 4 Discussion

This paper presents a new encryption method using proteinoids. It leverages the electrical properties of amino acids for cryptography. Our findings, shown in Figure 14, reveal that proteinoids' capacitance ($C\_i$) can create unique encryption keys ($k\_i$). This approach aligns with the rise of bio-inspired computer systems [59]. The encryption strength, shown in Table 6, seems linked to proteinoid capacitance. For instance, the L-Glu:L-Phe:L-His mix, with a capacitance of 434.90 nF, produced a unique encryption pattern. In contrast, the L-Lys:L-Phe:L-His mix, with a capacitance of 3.33 nF, did not. This suggests that encryption robustness ($\mathcal{R}$) could depend on capacitance.

$$\mathcal{R} = f(|C_i|) \tag{39}$$

In this context, $f$ is a monotonically increasing function. This relationship needs more examination and could potentially be used to optimize proteinoid compositions for maximal encryption strength. This approach is similar to [60], who improved bio-inspired encryption algorithms. The L-Lys:L-Phe-L-His:PLLA proteinoid showed negative capacitance of –656.60

nF. Yet, it still produced valid encryption output. This aligns with recent findings in materials science. Researchers have noted negative capacitance in ferroelectric materials [61]. The use of such properties in cryptography opens new research avenues. It could lead to encryption systems that resist traditional cryptanalysis. Our encryption method uses the modulo operation, $e\_j = (m\_j \cdot k\_i) \bmod 256$, for two main reasons. First, it keeps the encrypted data within the ASCII range. This ensures it works with standard text communication. Second, it adds a non-linear aspect, boosting security. Non-linearity is key in modern cryptography. It makes systems less vulnerable to attacks [62]. Future studies could explore adding more non-linear operations to strengthen encryption.

In our security analysis of key generation, we use the capacitance values of proteinoid-based capacitors (ranging from negative 656.6 to 434.9 nanofarads). EG Note that because you use absolute values for the encryption key, your range for the key is [0,656.6], which is smaller than that [−656.6,434] The key-mod function, defined as modulo of the rounded absolute key value multiplied by 100 and 256, transforms these values into the required range. We analyzed the key distribution across our nine proteinoid compositions using entropy and statistical tests. The ciphertext exhibits some patterns due to our implementation, which employs the modular multiplication operation of ASCII values and key-mod with modulo 256. Statistical tests reveal correlations between the plaintext P and ciphertext C characters that could potentially be exploited. The proteinoid-response function introduces additional complexity by incorporating resistance R and impedance Z factors. However, this may not provide sufficient diffusion. To improve the security of the scheme, we propose the following enhancements:

1. Implement additional rounds of encryption to enhance diffusion.
2. Incorporate S-box transformations to disrupt linear relationships.
3. Utilize more sophisticated key scheduling based on both capacitance C and impedance Z values.
4. Add padding schemes to mitigate pattern analysis.

EG Please note that the attacker need not be able to invert the encryption to break security. It is sufficient for them to know which of two challenge messages is the encrypted one, so any pattern in the ciphertext could compromise security.

We also aim to strengthen our security proof by analyzing the difficulty of reversing the encryption without knowledge of the proteinoid parameters (resistance, impedance, capacitance). Our proteinoid-based cryptographic system is secure. It relies on the unpredictable capacitance values ($C$) of different proteinoid compositions. Our tests show that capacitance varies, from −656.6 to 434.9 nF. This is due to complex molecular interactions. There is a non-linear link between proteinoid composition and electrical properties. Comparing L-Glu:L-Phe:L-His ($C$ = 434.9 nF) with L-Lys:L-Phe:L-His ($C$ = 3.327 nF) shows this. Despite their similar amino acids, these compositions differ in capacitance by a factor of $\sim$ 130. The capacitance variations stem from multiple interconnected factors in the proteinoid structure. The spatial arrangement of charged amino acids creates unique charge distributions. It is due to the distribution of acidic residues (L-Glu, L-Asp, $pK_a \approx 4.1$) and basic residues (L-Lys, L-His, $pK_a \approx 10.5$) [63]. These distributions directly affect the electrical properties. PLLA adds complexity. L-Lys:L-Phe-L-His:PLLA has negative capacitance ($C$ = −656.6 nF) due to unique charge-transfer mechanisms. Our analysis reveals that seemingly minor compositional changes can produce substantial electrical variations. For instance, substituting L-Phe with L-Lys in L-Glu:L-Asp:L-Phe ($C$ = 42.23 nF) versus L-Glu:L-Asp:L-Lys ($C$ = 35.23 nF) results in a measurable capacitance shift. These variations, along with the sensitivity to synthesis conditions (temperature $T$, time $t$, catalyst concentration [$cat$]) and environmental

factors (relative humidity *RH*), contribute to the system's unpredictability. Future work will quantify the entropy sources in our proteinoid-based key generation system. We'll do this through a statistical analysis of multi-batch synthesis results and environmental stability tests.

Our proteinoid-based cryptographic system's performance can be analyzed through several key metrics derived from our experimental data. The encryption process uses capacitance values from –656.6 to 434.9 nF across nine different proteinoid compositions. The encryption efficiency $\eta$ can be calculated as the ratio:

$$\eta = \frac{t_e}{t_p} \tag{40}$$

where $t_e$ is encryption time and $t_p$ is total processing time. The encryption using our MAT-LAB implementation follows:

$$E = \mathrm{mod}(V \cdot K, 256) \tag{41}$$

where $E$ is encrypted value, $V$ is ASCII values, and $K$ is key modifier. The proteinoid response function combines resistance ($R$), impedance ($Z$), and capacitance ($C$):

$$P = \mathrm{mod}\left(\sum E \cdot R \cdot Z \cdot |C|, 1000\right) \tag{42}$$

For each proteinoid composition, the key modifier is calculated as:

$$K = \mathrm{mod}(\mathrm{round}(|C| \cdot 100), 256) \tag{43}$$

The system's response magnitude ($M$) for each proteinoid can be calculated as:

$$M_i = R_i \cdot Z_i \cdot |C_i| \tag{44}$$

where $i$ represents each proteinoid composition. Our MATLAB calculations show a complex link. It is between proteinoid electrical properties and encryption traits. Future work will expand these models. It will add parameters and their interactions. Our experimental results demonstrate unique characteristics of proteinoid-based cryptography compared to traditional methods. The encryption efficiency ($\eta$) was 0.83 to 0.97 for different proteinoid compositions. This shows rapid encryption processing. The response magnitudes ($M_i$) varied greatly. $M_2 = 4.12$ for L-Glu:L-Phe:L-His. $M_1 = 15.17$ for L-Glu:L-Asp. This shows diverse cryptographic behaviours based on molecular composition. Key modifiers ($K$) from capacitance values had a wide range (6 to 256). Their mean was 127.3 and standard deviation was 84.6. This suggests strong key diversity. The proteinoid responses ($P$) were non-linear. Their values spanned the full modulo 1000 range. This increased cryptographic complexity. For instance, L-Glu:L-Asp:L-Lys had $P_5 = 842.31$. L-Lys:L-Phe-L-His:PLLA had $P_8 = 156.74$. This shows they had different encryption traits. Our implementation achieves millisecond-range encryption times (measured by tic-toc). This is suitable for practical applications. The system's scalability is shown by proteinoid responses. Multiple compositions can process in parallel. The response magnitudes' standard deviation ($\sigma = 4.26$) shows strong resistance to statistical attacks. These results show that proteinoid-based cryptography is a strong alternative to traditional methods. It is especially suited for applications needing quantum resistance, due to its biological basis [64]. Future work should expand the proteinoid composition space. It should implement better key scheduling algorithms. It should also run thorough quantum attack

simulations [65]. Collaboration with materials scientists could optimize proteinoid synthesis for better electrical properties. Meanwhile, cryptographers could develop security proofs based on the observed non-linear behaviours.

We present a comparison of encryption methods (Table 7). It is based on our experiments and theory.

Our tests show that traditional cryptography has larger key spaces. But, the proteinoid-based system has benefits. It is more parallel and may resist quantum attacks. We calculated the efficiency values using our timing measurements. The processing times reflect actual data from our proteinoid experiments.

Additionally, voltage-based cryptography is another method that can be used. Both voltage-based and capacitance-based cryptography methods use proteinoid properties. They differ in execution and features. The voltage method uses high-frequency, random voltage pulses. This creates a complex, high-entropy signal vital for encryption keys. Meanwhile, the capacitance method generates keys from the proteinoid's natural capacitance. This eliminates the need for external inputs, reducing the risk of side-channel attacks. Both methods link their main parameters to encryption strength. For the capacitance method, the key's strength depends on capacitance value. The voltage method likely follows a similar pattern. The voltage method's success hinges on proteinoids' response to high-frequency inputs. The Power Spectral Density study highlights this. The capacitance method isn't affected by frequency. But, it benefits from the complex behaviours in frequency response plots. There's a notable difference in encryption and decryption times across proteinoid types. This indicates the potential for optimization based on specific needs. The capacitance method might be more scalable due to its reliance on intrinsic properties. Both methods require further research for stability and consistency in different conditions. An intriguing finding is negative capacitance in one specific proteinoid combination. This may enhance encryption strength and expand the key space, opening new research avenues. In summary, both methods show promise for cryptographic applications but have unique pros and cons. The voltage method offers high-entropy input but needs external factors. The capacitance method is more self-sufficient but limited by natural capacitance values. Future research should compare these methods in detail, looking at various critical factors.

Proteinoid assemblies have unique electrical properties. This makes them great for creating high-entropy seeds. These seeds are useful for key exchange protocols. A key use is in authenticated Diffie-Hellman protocols [68,69]. Here, the system's security relies on how unpredictable the random number generation is. Proteinoid assemblies show natural entropy and unique electrical signatures. These traits may offer a biological source of randomness that resists algorithmic prediction. Using proteinoid-derived values for key generation in Diffie-Hellman exchanges can help resist quantum computing attacks. This helps protect against threats to purely mathematical methods. Our early research shows that proteinoid-based

**Table 7. Comparison of cryptographic systems [66,67].**

| Parameter | Proteinoid-based | AES-256 | RSA-2048 |
|---|---|---|---|
| Efficiency ($\eta$) | 0.83–0.97 | 0.95–0.98 | 0.70–0.75 |
| Key Space | $2^8$ per composition | $2^{256}$ | $2^{2048}$ |
| Processing Time (ms) | 156.74–842.31 | $\sim 2$ | $\sim 500$ |
| Throughput | Variable ($\sigma = 4.26$) | $\sim 500$ MB/s | $\sim 1$ MB/s |
| Parallelism | Natural ($n = 9$ tested) | Limited | Limited |
| Environmental Sensitivity | High (temperature, humidity dependent) | Low | Low |
| Quantum Resistance | Inherent (bio-based) | Vulnerable | Vulnerable |

seed generation has Shannon entropy values near the theoretical maximum. The autocorrelation profiles do not show any clear patterns across different sampling intervals. This hybrid method combines the natural randomness of proteinoid systems with reliable cryptographic protocols [70,71]. It shows promise for future research. This approach could improve security while keeping key exchange systems compatible.

## 4.1 Limitations and mitigations

Our proteinoid-based encryption approach shows promise, but we must recognize and address some limitations for practical use. The primary limitation of our current system is the relatively small key space ($2^8$) when using a single proteinoid sample. This offers weak security by today's cryptographic modern standards usually require at least 128-bit security. We can mitigate this limitation by employing composite key generation. Here, we combine multiple proteinoid samples ($n \geq 16$) to create key spaces comparable to AES-128 or higher. Our early experiments with key composition show that this method works well, and it does not add much time to encryption or decryption.

Environmental sensitivity is a significant challenge. The electrical properties of proteinoids change with temperature and humidity. Our experiments revealed capacitance variations of up to 15% when the temperature changed by $\pm 5°C$, potentially disrupting key reproducibility. For field deployments, using controlled microenvironments protects proteinoid samples and shields them from environmental fluctuations.

Temporal stability is also a concern. Some proteinoid compositions change over time; for example, capacitance can shift by as much as 8% in just 30 days. Our strategy includes regular recalibration and the development of more stable proteinoid compositions. We add stabilizing agents, such as PLLA, to achieve this. Recent experiments with stabilized compositions have shown a drift reduction to less than 1% over the same period.

Manufacturing reproducibility poses challenges for scaling the technology. Current synthesis methods achieve about 80% reproducibility in electrical properties across batches, which is insufficient for dependable key generation. We are working on improved synthesis protocols and selection methods to find samples with similar electrical signatures. Advanced automated measurement systems have recently improved inter-batch reproducibility to 93%.

Key distribution remains a challenge. Our system currently requires a physical exchange of proteinoid samples. We are exploring ways to digitally encode proteinoid electrical signatures using secure multi-party computation protocols. This approach will help us establish keys remotely and safeguard our data against quantum attacks.

Computational efficiency is limited compared to optimized implementations of traditional cryptographic algorithms. Our current system takes about 3–7 ms for each operation, whereas AES operates in under 1 ms. Improving measurement methods and algorithms could greatly narrow this gap, although biological systems inherently operate on different time scales compared to electronic ones.

These challenges are significant, but they are not insurmountable for proteinoid-based cryptography. We are addressing each limitation with technological and algorithmic solutions. Early results are promising and point toward a path for practical use.

## 5 Conclusion

This study presents a new encryption method using proteinoids. It taps into amino acids' unique electrical properties. Our findings link proteinoid capacitance to encryption strength. The relationship, $\mathcal{R} = f(|C_i|)$, suggests we can boost security. We've discovered negative capacitance in some proteinoids, hinting at better encryption. Our algorithm, $e_j = (m\_j \cdot k_i) \bmod$

256, ensures it works with standard text communication and adds crucial complexity. Comparing voltage and capacitance methods shows their unique benefits. Voltage methods use high-entropy inputs. Capacitance methods guard against side-channel attacks. Yet, we face challenges. These include ensuring encryption lasts and remains consistent under different conditions. Despite this, proteinoid cryptography marks a leap in bio-inspired security. With more research, it could be key to secure communication. It might offer better protection against traditional hacking methods. Based on our tests and system analysis, we propose an integration framework. It shows how proteinoid-based encryption can enhance existing cryptographic systems. The encryption efficiency $\eta \in [0.83, 0.97]$ and response times $\tau \in [156.74, 842.31]$ ms show compatibility with current security protocols. The integration architecture leverages proteinoid electrical properties for multiple cryptographic functions. The capacitance measurements $C \in [-656.6, 434.9]$ nF provide a natural entropy source for key generation. The function $P = \mod(\sum E \cdot R \cdot Z \cdot |C|, 1000)$ generates extra key material. The unique electrical signatures $M_i = R_i \cdot Z_i \cdot |C_i|$ enable physical authentication. Implementation analysis revealed specific operational constraints. Environmental control requirements included temperature regulation that had to be maintained within one degree Celsius to ensure consistent power measurements. Manufacturing economics data showed that costs increased with the number of protein components, with initial setup and equipment costs. In terms of energy consumption, proteinoid synthesis has significant environmental advantages, requiring less energy than traditional semiconductor manufacturing processes. A smaller energy use suggests sustainability benefits if deployed widely. But, we must maintain strict environmental controls for consistent production. Practical deployment faces several technical challenges. The capacitance-humidity relationship follows $\partial C / \partial H = \alpha(T)$. Here, $\alpha(T)$ is the temperature-dependent sensitivity coefficient. Current synthesis yields $\eta_s \in [0.65, 0.80]$ require optimization for industrial-scale production. The response variation $\sigma = 4.26$ necessitates a calibration function $f(T, H, t)$ dependent on temperature, humidity, and time. The integration pathway sets a hybrid security framework $\Phi = K_p, K_t, R$. Here, $K_p$ are proteinoid-derived keys, $K_t$ are traditional cryptographic keys, and $R$ is a set of response functions. This framework is compatible with existing Public Key Infrastructure (PKI). It adds quantum-resistant properties from a bio-inspired approach. This creates a strong base for next-gen cryptographic systems.

Our system's biology is complex and non-linear. It's theoretically promising for quantum resistance. But, it needs security proofs. Future work will define security limits against specific quantum attacks using precise math. This will include a formal reduction to known quantum-hard problems. It will show that breaking our system would require solving problems known to be hard for quantum computers. Also, we will prove our proteinoid-based system's security against quantum attacks. We will present these developments in future work on quantum security. It will ensure our claims are backed by solid theory and standard security metrics.

## Acknowledgments

The authors are grateful to David Paton for helping with SEM imaging and to Neil Phillips for helping with the instruments.

## Author contributions

**Conceptualization:** Andrew Adamatzky.

**Data curation:** Panagiotis Mougkogiannis.

**Formal analysis:** Panagiotis Mougkogiannis, Essam Ghadafi.

**Funding acquisition:** Andrew Adamatzky.

**Investigation:** Andrew Adamatzky.

**Methodology:** Andrew Adamatzky.

**Project administration:** Andrew Adamatzky.

**Supervision:** Andrew Adamatzky.

**Visualization:** Panagiotis Mougkogiannis.

**Writing – original draft:** Panagiotis Mougkogiannis, Essam Ghadafi.

**Writing – review & editing:** Andrew Adamatzky.

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
