## [Decision Letter · Decision Letter 0]

28 Nov 2024

PONE-D-24-49897Bio-Inspired Cryptography Based on Proteinoid AssembliesPLOS ONE

Dear Dr. Mougkogiannis,

Thank you for submitting your manuscript to PLOS ONE. After careful consideration, we feel that it has merit but does not fully meet PLOS ONE’s publication criteria as it currently stands. Therefore, we invite you to submit a revised version of the manuscript that addresses the points raised during the review process.

We look forward to receiving your revised manuscript.

Kind regards,

Daniel Ioan Hunyadi, Ph.D.

Academic Editor

PLOS ONE

Journal Requirements:

3. Thank you for stating the following financial disclosure: [y EPSRC Grant EP/W010887/1 “Computing with proteinoids”]. Please state what role the funders took in the study. If the funders had no role, please state: "The funders had no role in study design, data collection and analysis, decision to publish, or preparation of the manuscript." If this statement is not correct you must amend it as needed. Please include this amended Role of Funder statement in your cover letter; we will change the online submission form on your behalf.

4. Please expand the acronym “EPSRC” (as indicated in your financial disclosure) so that it states the name of your funders in full. This information should be included in your cover letter; we will change the online submission form on your behalf.

5. Please amend your list of authors on the manuscript to ensure that each author is linked to an affiliation. Authors’ affiliations should reflect the institution where the work was done (if authors moved subsequently, you can also list the new affiliation stating “current affiliation:….” as necessary).

Reviewers' comments:

Reviewer's Responses to Questions

**Comments to the Author**

1. Is the manuscript technically sound, and do the data support the conclusions?

Reviewer #1: Partly

Reviewer #2: Yes

2. Has the statistical analysis been performed appropriately and rigorously? 

Reviewer #1: No

Reviewer #2: I Don't Know

3. Have the authors made all data underlying the findings in their manuscript fully available?

Reviewer #1: No

Reviewer #2: Yes

4. Is the manuscript presented in an intelligible fashion and written in standard English?

Reviewer #1: Yes

Reviewer #2: Yes

5. Review Comments to the Author

Reviewer #1: Title- Bio-Inspired Cryptography Based on Proteinoid Assemblies

1. Justify the statement " Bioinspired cryptography has emerged as a significant area of research,".

2. The table 1 compare the bio inspired cryptography and key based cryptography, what is the need of both of the approach.

3. materials and methods section is too small, i have a suggestion to add more data from the result section.

Reviewer #2: This paper seems to have some innovative scientific crypto contribution benefiting from deterministic interactions of proteinoid assemblies. The work generates secure encryption keys for algorithms using unpredictability of proteinoid behaviour that is considered adaptable encryption framework resisting known cryptanalytic attacks. The exploration illustrates the implementation of bio-inspired system based on simulated proteinoid interactions resulting promissing encryption strength. The research show random key generation presenting security of standard statistical tests immune to classical and quantum computers attacks. This study innovates security via biological principles of potential processes in cryptography needing overall secrecy reconsideration in order to be in ready and complete. Therefore, the following secrecy references are recommended to be involved:

== “AI Based Mobile Edge Computing for IoT: Applications, Challenges, and Future Scope”, Arabian Journal for Science and Engineering (2022). http://doi.org/10.1007/s13369-021-06348-2

== "Personal Privacy Evaluation of Smart Devices Applications Serving Hajj and Umrah Rituals", Journal of Engineering Research (2021). http://doi.org/10.36909/jer.13199

== "Cybercrime on Transportation Airline", Journal of Forensic Research, ISSN: 2157-7145, 10(4):449 (2019)

== "Sensing e-Banking Cybercrimes Vulnerabilities via Smart Information Sciences Strategies", RAS Engineering and Technology 1(1):1-9 (2020)

== "Engineering Graphical Captcha and AES Crypto Hash Functions for Secure Online Authentication", Journal of Engineering Research (2021). http://doi.org/10.36909/jer.13761

The work is more interesting if considering the following image hiding strategies proposed within below encryption researches:

== "Varying PRNG to improve image cryptography implementation", Journal of Engineering Research 9(3A):153-183 (2021).

== "Remodeling Randomness Prioritization to Boost-up Security of RGB Image Encryption", Multimedia Tools and Applications (MTAP) 80(18):28521–28581 (2021)

== “Dynamic smart random preference for higher medical image confidentiality”, Journal of Engineering Research (2022) http://doi.org/10.36909/jer.17853

== "Improving data hiding within colour images using hue component of HSV colour space", CAAI Transactions on Intelligence Technology, 7(1): 56–68 (2022)

== "Efficient Image Reversible Data Hiding Technique Based on Interpolation Optimization", Arabian Journal for Science and Engineering (AJSE) 46(9):8441–8456 (2021)

== "Efficient Reversible Data Hiding Multimedia Technique Based on Smart Image Interpolation", Multimedia Tools and Applications (MTAP) 79(39):30087-30109 (2020)

6. PLOS authors have the option to publish the peer review history of their article (what does this mean?). If published, this will include your full peer review and any attached files.

Reviewer #1: **Yes: **Sultan H. Almotiri

Reviewer #2: No

---

## [Author Response · Author response to Decision Letter 1]

2 Dec 2024

response to reviewers letter attached!

---

## [Decision Letter · Decision Letter 1]

14 Jan 2025

PONE-D-24-49897R1Bio-Inspired Cryptography Based on Proteinoid AssembliesPLOS ONE

Dear Dr. Mougkogiannis,

Thank you for submitting your manuscript to PLOS ONE. After careful consideration, we feel that it has merit but does not fully meet PLOS ONE’s publication criteria as it currently stands. Therefore, we invite you to submit a revised version of the manuscript that addresses the points raised during the review process.

We look forward to receiving your revised manuscript.

Kind regards,

Daniel Ioan Hunyadi, Ph.D.

Academic Editor

PLOS ONE

Journal Requirements:

Reviewers' comments:

Reviewer's Responses to Questions

**Comments to the Author**

1. If the authors have adequately addressed your comments raised in a previous round of review and you feel that this manuscript is now acceptable for publication, you may indicate that here to bypass the “Comments to the Author” section, enter your conflict of interest statement in the “Confidential to Editor” section, and submit your "Accept" recommendation.

Reviewer #3: All comments have been addressed

Reviewer #4: (No Response)

Reviewer #5: (No Response)

2. Is the manuscript technically sound, and do the data support the conclusions?

Reviewer #3: Yes

Reviewer #4: Yes

Reviewer #5: Yes

3. Has the statistical analysis been performed appropriately and rigorously? 

Reviewer #3: N/A

Reviewer #4: Yes

Reviewer #5: Yes

4. Have the authors made all data underlying the findings in their manuscript fully available?

Reviewer #3: No

Reviewer #4: No

Reviewer #5: Yes

5. Is the manuscript presented in an intelligible fashion and written in standard English?

Reviewer #3: Yes

Reviewer #4: Yes

Reviewer #5: Yes

6. Review Comments to the Author

Reviewer #3: This paper proposed a new cryptographic method by taking advantage of the unpredictability of protein-like behavior. It used simulated protein-like interactions to realize a bionic system, and the results met the current encryption strength standards and also had advantages in key generation and distribution. This paper has a clear structure and complete content, but there are still the following problems.

1�� The cryptographic primitive proposed in this paper is innovative, but lacks theoretical proof of its security. This paper proposed that the security of the cryptographic algorithm depends on the unpredictability of the key generated based on protein-like capacitors. However, the ciphertext after encoding and encryption still has certain statistical characteristics. The security of this part lacks corresponding proof and analysis.

2�� There is a lack of evidence and analysis on the unpredictability of keys generated based on protein-like capacitors. For example, do different protein-like proteins have similar capacitance values? Is there a relatively fixed numerical relationship between capacitance and the chemical composition, structural characteristics, and polymerization mode of protein-like proteins?

3�� There is a problem with the image layout, and the corresponding image cannot be indexed.

Reviewer #4: (No Response)

Reviewer #5: Provide more detailed explanations of how proteinoid assemblies specifically interact with cryptographic algorithms.

This would make the biological inspiration clearer and more accessible for readers unfamiliar with proteinoids.

Include a comparison table between proteinoid-based cryptography and traditional encryption methods in terms of efficiency, security, and scalability. This could highlight the advantages and unique aspects of the bio-inspired approach.

Explore ways to optimize the proteinoid-based encryption system for real-time applications.

Include a broader range of test cases and attack simulations, especially those involving different types of quantum and classical attacks, to strengthen the claim of robustness.

Suggest the potential for further collaboration with biologists, material scientists, and cryptographers to refine proteinoid-based systems and explore additional biological processes for future cryptographic improvements.

All figures looks blurry and not clearly visible to depict the findings.

Discuss how the proteinoid-based encryption can be integrated into existing security frameworks or protocols to assess compatibility and ease of adoption in current cryptographic infrastructures.

Address the potential environmental and cost benefits or challenges in producing proteinoids for widespread use in cryptography.

Proof read the paper and correct grammatical errors through out the work.

7. PLOS authors have the option to publish the peer review history of their article (what does this mean?). If published, this will include your full peer review and any attached files.

Reviewer #3: No

Reviewer #4: No

Reviewer #5: No

---

## [Author Response · Author response to Decision Letter 2]

20 Jan 2025

Dear Editor,

The response to the reviewers is attached as a PDF file.

Thanks,

The authors

---

## [Decision Letter · Decision Letter 2]

9 Feb 2025

PONE-D-24-49897R2Bio-Inspired Cryptography Based on Proteinoid AssembliesPLOS ONE

Dear Dr. Mougkogiannis,

Thank you for submitting your manuscript to PLOS ONE. After careful consideration, we feel that it has merit but does not fully meet PLOS ONE’s publication criteria as it currently stands. Therefore, we invite you to submit a revised version of the manuscript that addresses the points raised during the review process.

We look forward to receiving your revised manuscript.

Kind regards,

Daniel Ioan Hunyadi, Ph.D.

Academic Editor

PLOS ONE

Reviewers' comments:

Reviewer's Responses to Questions

**Comments to the Author**

1. If the authors have adequately addressed your comments raised in a previous round of review and you feel that this manuscript is now acceptable for publication, you may indicate that here to bypass the “Comments to the Author” section, enter your conflict of interest statement in the “Confidential to Editor” section, and submit your "Accept" recommendation.

Reviewer #6: All comments have been addressed

Reviewer #7: All comments have been addressed

2. Is the manuscript technically sound, and do the data support the conclusions?

Reviewer #6: No

Reviewer #7: No

3. Has the statistical analysis been performed appropriately and rigorously? 

Reviewer #6: No

Reviewer #7: No

4. Have the authors made all data underlying the findings in their manuscript fully available?

Reviewer #6: No

Reviewer #7: No

5. Is the manuscript presented in an intelligible fashion and written in standard English?

Reviewer #6: No

Reviewer #7: No

6. Review Comments to the Author

Reviewer #6: In principle, a process that is not replicable in the same way, with the same values is random and one that generates uncorrelated output values, but which can be replicated by knowing some parameters of the generation algorithm, will produce a pseudorandom set. From the description made, the primary model from which the authors start could be one used for random systems. The applicability would be limited by the transmission model of these data sets, called seeds in cryptography, and used within pseudorandom models, as initial values of the system.

To get out of this scope of applicability, which does not allow the correlation of the emitter and the receiver in order to fulfill the fundamental axiom of a cryptographic system, the authors should illustrate the completeness of the proposed model. In this sense, it would be useful:

1. A dedicated section to clearly illustrate how the correlation of keys from the sender of the encryption data system to the receiver of the encrypted data is achieved, in order to fulfill d(e(m))=m?

2. A clear and mathematically substantiated explanation for a correlative question with the applicability of the proposed solution: why do the electrical properties of proteinoids provide mathematical security comparable to traditional methods (AES, RSA, etc.)?

3. How is it ensured that an attacker cannot replicate a set of proteinoids and generate the same keys?

4. The key generation algorithm is based on measurable physical values, but experimental variability could affect the reproducibility of the keys. How is the synchronization of the two parties involved in communication achieved? And in this sense, the correlation of these explanations with point 3.

Reviewer #7: A process that cannot be reproduced in the same way and with the same initial values can be considered random, since it does not show determinism in its evolution. In contrast, a system that generates uncorrelated output values, but which can be replicated by knowing specific parameters of the generation algorithm, produces a pseudorandom sequence. From the description provided, the fundamental model proposed by the authors seems to be usable in systems with a random character. However, its applicability could be limited by the mechanism for distributing these data sets, called seeds in cryptography, essential in pseudorandom algorithms, where they are used as initial values to determine the behavior of the system. If the authors want to choose this way of generating cryptographic primitives, perhaps a study that would use the values generated for the seed of models derived from authenticated Diffie-Helmann would be more useful.

To clarify the sustainability of the proposed solution, especially in the context where, if functional, the proposed method would be an idea that should revolutionize the way cryptographic systems can be built, it would be useful to consider the following:

1. A dedicated section describing the limitations of the method and describing possible mitigation methods.

2. Table 1 provides a comparison between different cryptographic technologies; however, for a more rigorous evaluation, it would be necessary to include quantitative criteria, such as system performance, security level, computational complexity or energy efficiency.

3. Clarify and rigorously detail the mechanisms by which proteinoids constitute a viable cryptographic solution, highlighting the aspects that give them security, robustness and practical applicability compared to traditional cryptographic methods.

4. Provide a detailed and rigorous analysis of the entropy and security level of the generated keys, highlighting the degree of randomization, resistance to cryptanalytic attacks and their compliance with current security standards.

5. A review of the abstract and introduction would be useful to explain more clearly, beyond a narrative presentation, how the proposed solution is built and how it works.

6. The completeness and synchronization of the key models and algorithms used by the sender and receiver.

7. PLOS authors have the option to publish the peer review history of their article (what does this mean?). If published, this will include your full peer review and any attached files.

Reviewer #6: No

Reviewer #7: No

---

## [Author Response · Author response to Decision Letter 3]

26 Mar 2025

Dear Editor

The response to reviewers document has been attached in previous step.

Thanks

The Authors

---

## [Decision Letter · Decision Letter 3]

2 May 2025

Bio-Inspired Cryptography Based on Proteinoid Assemblies

PONE-D-24-49897R3

Dear Dr. Mougkogiannis,

We’re pleased to inform you that your manuscript has been judged scientifically suitable for publication and will be formally accepted for publication once it meets all outstanding technical requirements.

Kind regards,

Daniel Ioan Hunyadi, Ph.D.

Academic Editor

PLOS ONE

Additional Editor Comments (optional):

Reviewers' comments:

Reviewer's Responses to Questions

**Comments to the Author**

1. If the authors have adequately addressed your comments raised in a previous round of review and you feel that this manuscript is now acceptable for publication, you may indicate that here to bypass the “Comments to the Author” section, enter your conflict of interest statement in the “Confidential to Editor” section, and submit your "Accept" recommendation.

Reviewer #6: All comments have been addressed

Reviewer #8: All comments have been addressed

2. Is the manuscript technically sound, and do the data support the conclusions?

Reviewer #6: Yes

Reviewer #8: Yes

3. Has the statistical analysis been performed appropriately and rigorously? 

Reviewer #6: Yes

Reviewer #8: Yes

4. Have the authors made all data underlying the findings in their manuscript fully available?

Reviewer #6: No

Reviewer #8: Yes

5. Is the manuscript presented in an intelligible fashion and written in standard English?

Reviewer #6: Yes

Reviewer #8: Yes

6. Review Comments to the Author

Reviewer #6: The current version of the article deals with the subject addressed in a clear way and has responded fairly enough to all the comments made.

In this form the article describes clearly enough the proposed cryptographic technique. Even if the real applicability of the proposed solution has certain functional implementation problems, from a purely theoretical point of view of the study this can be considered a research article.

Reviewer #8: The authors have significantly improved the manuscript over the course of the three revisions. The final version presents a compelling integration of biological principles with cryptographic innovation, specifically through the use of proteinoid assemblies. The experimental design has become clearer, and the encryption methodology—particularly the translation of electrical capacitance into secure key generation—is now articulated with better precision. The statistical validation of key randomness, the complexity analysis, and the energy efficiency metrics help substantiate the claims made. Some minor linguistic polishing could further improve the readability of the paper, but overall, the manuscript has matured into a coherent and original contribution to the field of bio-inspired cryptography. I recommend it for publication.

7. PLOS authors have the option to publish the peer review history of their article (what does this mean?). If published, this will include your full peer review and any attached files.

Reviewer #6: No

Reviewer #8: **Yes: **AMJED ABBAS

---

## [Editor Report · Acceptance letter]

PONE-D-24-49897R3

PLOS ONE

Dear Dr. Mougkogiannis,

I'm pleased to inform you that your manuscript has been deemed suitable for publication in PLOS ONE. Congratulations! Your manuscript is now being handed over to our production team.

Kind regards,

on behalf of

Dr. Daniel Ioan Hunyadi

Academic Editor

PLOS ONE